# Unbalanced emission reductions of different species and sectors in China during COVID-19 lockdown derived by multi-species surface observation assimilation

Lei Kong[1,3], Xiao Tang[*,1,3], Jiang Zhu[2,3], Zifa Wang[1,3,4], Yele Sun[1,3], Pingqing Fu[5], Meng Gao[6], Huangjian Wu[1,3], Miaomiao Lu[7], Qian Wu[1,3], Shuyuan Huang[8], Wenxuan Sui[1], Jie Li[1,3], Xiaole Pan[1,3], Lin Wu[1,3], Hajime Akimoto[9], Gregory R. Carmichael[10]

[1]State Key Laboratory of Atmospheric Boundary Layer Physics and Atmospheric Chemistry (LAPC), Institute of Atmospheric Physics, Chinese Academy of Sciences, Beijing 100029, China
[2]CAS-TWAS Center of Excellence for Climate and Environment Sciences (ICCES), Institute of Atmospheric Physics, Chinese Academy of Sciences, Beijing 100029, China
[3]University of Chinese Academy of Sciences, Beijing 100049, China
[4]Center for Excellence in Regional Atmospheric Environment, Institute of Urban Environment, Chinese Academy of Sciences, Xiamen 361021, China
[5]Institute of Surface-Earth System Science, Tianjin University, Tianjin 300072, China
[6]Department of Geography, State Key Laboratory of Environmental and Biological Analysis, Hong Kong Baptist University, Hong Kong SAR, China
[7]State Environmental Protection Key Laboratory of Urban Ambient Air Particulate Matter Pollution Prevention and Control, College of Environmental Science and Engineering, Nankai University, Tianjin 300350, China
[8]Chengdu University of Information Technology, Chengdu 610225, China
[9]National Institute for Environmental Studies, Onogawa, Tsukuba 305-8506, Japan
[10]Center for Global and Regional Environmental Research, University of Iowa, Iowa City, IA 52242, USA

*Correspondence to*: Xiao Tang (tangxiao@mail.iap.ac.cn)

**Abstract.** The unprecedented lockdown of human activities during the COVID-19 pandemic have significantly influenced the social life in China. However, understanding of the impact of this unique event on the emissions of different species is still insufficient, prohibiting the proper assessment of the environmental impacts of COVID-19 restrictions. Here we developed a multi-air pollutant inversion system to simultaneously estimate the emissions of $NO_x$, $SO_2$, CO, $PM_{2.5}$ and $PM_{10}$ in China during COVID-19 restrictions with high temporal (daily) and horizontal (15km) resolutions. Subsequently, contributions of emission changes versus meteorology variations during COVID-19 lockdown were separated and quantified. The results demonstrated that the inversion system effectively reproduced the actual emission variations of multi-air pollutants in China during different periods of COVID-19 lockdown, which indicate that the lockdown is largely a nationwide road traffic control measure with $NO_x$ emissions decreased substantially by ~40%. However, emissions of other air pollutants were found only decreased by ~10%, because power generation and heavy industrial processes were not halted during lockdown, and residential activities may actually have increased due to the stay-at-home orders. Consequently, although obvious reductions of $PM_{2.5}$ concentrations occurred over North China Plain (NCP) during lockdown period, the emission change only accounted for 8.6% of $PM_{2.5}$ reductions, and even led to substantial increases of $O_3$. The meteorological variation instead dominated the changes in $PM_{2.5}$ concentrations over NCP, which contributed 90% of the $PM_{2.5}$ reductions over most parts of NCP region. Meanwhile,

our results suggest that the local stagnant meteorological conditions together with inefficient reductions in $PM_{2.5}$ emissions
were the main drivers of the unexpected $PM_{2.5}$ pollution in Beijing during lockdown period. These results highlighted that
traffic control as a separate pollution control measure has limited effects on the coordinated control of $O_3$ and $PM_{2.5}$
concentrations under current complex air pollution conditions in China. More comprehensive and balanced regulations for
multiple precursors from different sectors are required to address $O_3$ and $PM_{2.5}$ pollution in China.

## 1 Introduction

A novel coronavirus disease (COVID-19) broke out in Wuhan at the end of 2019 but quickly spread across the whole
China within a month. To curb the spread of the virus, strict epidemic control measures were implemented by Chinese
governments to prevent large gatherings, including strict travel restriction, shutting down of non-essential industries, extended
holidays, closing of schools and entertainment houses (Cheng et al., 2020). These restrictions have had a significant impact on
the industrial activities and social life, as exemplified by the drop of China's industrial output by 15-30%
(https://data.stats.gov.cn/, last accessed on 22 Oct, 2022) and the dramatic decrease of traffic flow by 60–90% in major cities
of China during COVID-19 epidemic (http://jiaotong.baidu.com/, last accessed on 22 Oct, 2022), which provides us a natural
experiment to examine the responses of the emissions and air quality on the changes in human activities.
It has been well documented that the short-term stringent emission control targeted on power generator or heavy industry
enacted by Chinese government during certain societal events, such as the 2008 Olympics Games, 2014 Asia-Pacific Economic
Cooperation conference and 2015 China Victory Day Parade, is an effective way to reduce emissions and improve air quality
(Okuda et al., 2011; Wang et al., 2014; Tang et al., 2015; Zhang et al., 2016; Wu et al., 2020; Chu et al., 2018). However,
different from those stringent emission controls, the COVID-19 restrictions are inclined to affect emissions from sectors more
closely to social life whose influence on emissions has still not well been assessed. Previous studies suggest that the COVID-
19 restrictions have substantially reduced the China's anthropogenic emissions from almost all sectors (Zheng et al., 2021;
Huang et al., 2021; Xing et al., 2020). For example, by using a bottom-up method based on near-real-time activity data, Zheng
et al. (2021) reported that the emissions of $NO_x$, $SO_2$, CO and primary $PM_{2.5}$ decreased by 36%, 27%, 28% and 24% during
COVID-19 restrictions, mostly due to the reductions in industry and transportation sector. Xing et al. (2020), by using a
response model, estimated stronger COVID-19 shutdown effects on emissions over the North China Plain (NCP) with
emissions of $NO_x$, $SO_2$ and primary $PM_{2.5}$ dropped by 51%, 28% and 63%, respectively. Others argue that the COVID-19
restriction may mainly affect the emissions from transportation, light industry and manufacturing, while it has much smaller
effects on the emissions from the power generator and heavy industry because of their non-interruptible processes (Chu et al.,
2021; Hammer et al., 2021; Le et al., 2020; Zhao et al., 2020). Moreover, the residential emissions may even increase during
the COVID-19 lockdown due to the increased demanding for space heating and cooking with the stay-at-home orders.
Therefore, Le et al. (2020) only considered the $NO_x$ reductions during COVID-19 restrictions in their investigation of the
severe haze during COVID-19 lockdown, and similarly, Hammer et al. (2021) only considered the emission reductions in the
transportation sector. This indicates that there has large uncertainty in the current understanding of the effects of COVID-19
restrictions on the emissions of different species.

71       Quantification of the emission changes of different species and different sectors during the COVID-19 lockdown is thus
necessary for the comprehensive understanding of the environmental impacts of COVID-19 restrictions. In particular, although
observations indeed show decreases of air pollutant concentrations during COVID-19 restrictions (Fan et al., 2020; Wang et
al., 2021; He et al., 2020; Shi and Brasseur, 2020), the air quality improvement is much smaller than the expected (Shi et al.,
2021; Diamond and Wood, 2020; Yan et al., 2022). Moreover, severe haze still occurred in northern China (Sulaymon et al.,
2021; Le et al., 2020) and $O_3$ concentrations even showed significant increases (Zhang et al., 2021; Li et al., 2020). A number
of studies were conducted to explain this anomalistic air quality change by analyzing the effects of emission changes,
meteorological variations and secondary production (Huang et al., 2021; Le et al., 2020; Hammer et al., 2021; Zhao et al.,
2020; Zhao et al., 2021; Sulaymon et al., 2021; Wang et al., 2020; Li et al., 2021). However, due to the unknown emission
changes during COVID-19 restrictions, the emission reduction scenarios that used to represent the COVID-19 shutdown effects
varied among different studies and did not consider the spatial and temporal heterogeneity of the emission changes, leading to
biases in the model simulation (Zhao et al., 2021; Li et al., 2021; Hammer et al., 2021; Zheng et al., 2021) and uncertainty in
the quantification of the contributions of different factors.

84       Pioneer studies by Zheng et al. (2021) and Forster et al. (2020) have derived multi-air pollutant emissions from social
activity data using a bottom-up method, but due to the lack of detailed social activity data, large uncertainties existed in their
estimates. The meteorologically and seasonally driven variability of the concentrations of air pollutants also prohibit drawing
fully quantitative conclusions on the changes of emissions based on observations alone (Levelt et al., 2022). The emission
inversion technique, which takes advantage of the chemical transport model (CTM) and real-time observations, provides an
attractive way to estimate the sector-specific and space-based emission changes during COVID-19 restrictions, as shown in
Zhang et al. (2020), Zhang et al. (2021), Feng et al. (2020) and Hu et al. (2022). However, these studies only inversed the
emissions of single species (e.g., $NO_x$ and $SO_2$) without insights into multiple species. In view of this discrepancy, in this study
we developed a multi-air pollutant inversion system to simultaneously estimated the multi-air pollutant emissions in China,
including $NO_x$, $SO_2$, CO, $PM_{2.5}$ and $PM_{10}$, during the COVID-19 restrictions using an ensemble Kalman filter (EnKF) and
surface observations from the China National Environmental Monitoring Centre (CNEMC). Subsequently, the inversed
emission inventory was used to quantify the contributions of emission changes versus meteorology variations to the changes
in $PM_{2.5}$ and $O_3$ concentrations over the NCP region during the COVID-19 restrictions.
**2 Method and data**

98       We developed a high-resolution multi-air pollutant inversion system to estimate the daily emissions of $NO_x$, $SO_2$, CO,
$PM_{2.5}$ and $PM_{10}$ in China from 1 Jan to 29 Feb 2020 when the COVID-19 pandemic was at its most serious and the effects of
the COVID-19 restrictions were most profound in China. This system uses the NAQPMS (Nested Air Quality Prediction
Modelling System) model as the forecast model and the EnKF coupled with the state argumentation method as the inversion
method. It has the capabilities of simultaneous inversion of multi-air pollutant emissions at high temporal (daily) and spatial
(15km) resolutions. An iteration inversion scheme was also developed in this study to address the large biases in the a priori
emissions. In order to better characterize the emission changes during the COVID-19 restrictions, the whole time period was
divided into three periods according to different control phases of COVID-19 and the timing of the Chinese Lunar New Year:
before lockdown (P1, January 1-20), lockdown (P2, January 21-February 9) and after back-to-work day (P3, February 10-29).
Emission changes in different regions of China were also analyzed, including the North China Plain (NCP), Northeast China
(NE), Southeast China (SE), Southwest China (SW), Northwest China (NW) and Central regions (defined in Fig. 1) to
investigate the responses of emissions to the COVID-19 restrictions in different regions. In the following sections, we briefly
introduce each component of the inversion system.

**2.1 Chemical transport model and its configuration**

The NAQPMS model was used as the forecast model to represent the atmospheric chemistry in this study, which has been
used in previous inversion studies (Tang et al., 2011; Tang et al., 2013; Kong et al., 2019; Wu et al., 2020), where detailed
descriptions of NAQPMS are available. The Weather Research and Forecasting Model (WRF)(Skamarock, 2008) is used to
provide the meteorological inputs to the NAQPMS model.
Figure 1 shows the modelling domain of this study with a high horizontal resolution of 15 km. The a priori emission
inventory used in this study includes monthly anthropogenic emissions from the HTAP_v2.2 emission inventory for the base
year of 2010 (Janssens-Maenhout et al., 2015), biomass burning emissions from the Global Fire Emissions Data base (GFED)
version 4 (Randerson et al., 2017; Van Der Werf et al., 2010), biogenic volatile organic compound (BVOC) emissions from
MEGAN-MACC (Sindelarova et al., 2014), marine volatile organic compound emissions from the POET database (Granier et
al., 2005), soil $NO_x$ emissions from the Regional Emission inventory in Asia (Yan et al., 2003) and lightning $NO_x$ emissions
from Price et al. (1997). Chemical top and boundary conditions were provided by the global CTM MOZART (Model for
Ozone and Related Chemical Tracers) (Brasseur et al., 1998; Hauglustaine et al., 1998). We assumed no monthly variations in
the a priori emission inventory and used January's emission inventory for the whole simulation period so that the emission
variation was solely derived from the surface observations. A two-week free run of NAQPMS was conducted as a spin-up
time. For each day's meteorological simulation, a 36-h free run of WRF was conducted, of which the first 12-h simulation was
a spin-up run and the next 24-h simulation provided the meteorological inputs to NAQPMS. Initial and boundary conditions
for the meteorological simulation were provided by the National Center for Atmospheric Research/National Center for
Environment Prediction (NCAR/NCEP) 1° ×1° reanalysis data. Evaluation results for the WRF simulation are available in
Text S1 in Supplement.

## 2.2 Surface Observations

The hourly concentrations of $NO_2$, $SO_2$, CO, $PM_{2.5}$ and $PM_{10}$ from CNEMC were used in this study to estimate the emissions during COVID-19. The spatial distributions of these observation sites are shown in Fig. 1, which contains 1436 observation sites covering most regions of China. Before assimilation, outliers of observations were first filtered out using the automatic outlier detection method developed by Wu et al. (2018) to prevent the adverse effects of the outliers on data assimilation. Then, the hourly concentrations were averaged to the daily values for the inversions of daily emissions.

The observation error is one of the key inputs to the data assimilation, which together with the background error determine the relative weights of the observation and background values on the analysis. The observation error includes measurement error and representativeness error. The measurement error of each species was designated according to the officially released documents of the Chinese Ministry of Ecology and Environmental Protection (HJ 193-2013 and HJ 654-2013, available at http://www.cnemc.cn/jcgf/dqhj/, last accessed on 22 Oct 2022), which is 5% for $PM_{2.5}$ and $PM_{10}$ and 2% for $SO_2$, $NO_2$ and CO. A representativeness error arises from the different spatial scales that the discrete observation data and model simulation represent, which was estimated based on the previous study by Li et al. (2019) and Kong et al. (2021). It should be noted that the $NO_2$ measurement from CNEMC is made by the chemiluminescent analyser with a molybdenum converter. Due to the interference of $HNO_3$, PAN and alkyl nitrates (AN), the $NO_2$ concentrations can be overestimated (Dunlea et al., 2007; Lamsal et al., 2008) that may lead to spurious decreases in $NO_x$ emissions during the lockdown period. Previous studies usually use chemical transport model to simulate $NO_x$, $HNO_3$, PAN and AN to produce correction factors (CFs) for the $NO_2$ measurements (Cooper et al., 2020; He et al., 2022) using the following relationship proposed by Lamsal et al. (2008):

$$CF = \frac{[NO_2]}{[NO_2]+0.95[PAN]+0.35[HNO_3]+\sum[AN]} \tag{1}$$

but the calculation of CF could be affected by the simulation errors in the model caused by uncertainties in emission inventory or other error sources, which may contaminate the observations. Therefore, similar to Feng et al. (2020), we did not correct the $NO_2$ measurement in our inversion of $NO_x$ emissions since there were large uncertainties in the $NO_x$ emissions during the COVID-19 pandemic that possibly led to erroneous CF. Since the EnKF considered the errors in observations through the use of observation error covariance matrix, the chemiluminescence monitor interference to $NO_2$ measurement were treated as the observation error during the assimilation. A sensitivity inversion experiment was also conducted based on the corrected $NO_2$ measurement using CF, which suggests that the chemiluminescence monitor interference only have small impacts on the inversed $NO_x$ emission in terms of magnitude and its variation during COVID-19 pandemic. Detailed results of the sensitivity experiment are available in Text S2 in Supplement.

## 2.3 Inversion estimation scheme

The EnKF coupled with the state augmentation method was used in this study to constrain the emissions of multiple species. EnKF is an advanced data assimilation method proposed by Evensen (1994) that features representation of the uncertainties of the model state by a stochastic ensemble of model realizations. Different from the mass balance method used

in Zhang et al. (2020) and Zhang et al. (2021) that has difficulties in accounting for nonlinear relationship between emissions
and concentrations and is more suitable for short-lived species (e.g. $NO_x$) under relatively coarse (>1°) resolutions (Streets et
al., 2013), the EnKF can consider the indirect relationship between emissions and concentrations caused by complex physical
and chemical processes in the atmosphere through the use of flow-dependent background error covariance produced by
ensemble CTM forecasts (Evensen, 2009; Miyazaki et al., 2012). Compared with the four-dimensional variational assimilation
method used in Hu et al. (2022), the EnKF method has comparable computational cost (Skachko et al., 2014) but is more easily
implemented without the need to develop complicated adjoint models for complex CTMs. The state augmentation method is
a commonly used parameter estimation method (Tandeo et al., 2020), in which the emissions of multi species are treated as
state variable and are simultaneously updated according to the relationship between the emissions and concentrations of related
species. Due to the chemical reactions in the atmosphere, the concentrations of different species are interrelated with each
other. For example, the ambient $PM_{2.5}$ is not only primarily emitted, but also formed secondarily through reactions with several
gaseous precursors, such as $NO_2$ and $SO_2$. This means that the estimations of $PM_{2.5}$ emission by single inversed estimation
method could be biased if the errors in $NO_2$ and $SO_2$ emissions were not corrected synchronously. Therefore, it is beneficial
to do the multi-species inversion estimation which can provide more constraints on the atmospheric chemical system and lead
to more reasonable inversion results. Meanwhile, the use of EnKF method coupled with the state augmentation method allows
the estimations of multi-species emissions almost without additional computational cost.
Appropriate estimation of the uncertainty in emissions and chemical concentrations is important for the performance of
inversion estimation using EnKF. Since the source emission data over mainland China in HTAP_v2.2 inventory is obtained
from the MIX inventory (Li et al., 2017b) , the uncertainties of emissions of different species, including PMF, PMC, BC, OC,
$NO_x$, CO, $SO_2$, $NH_3$ and NMVOC (nonmethane volatile organic compounds), were obtained from Li et al. (2017b) and Streets
et al. (2003), which were represented by an ensemble of perturbed emissions generated by multiplying the a priori emissions
with a perturbation factor $\beta_{i,s}$:
$$E_{i,s} = \beta_{i,s} \circ E_s^p, \; i = 1,2, \cdots N_{ens} \tag{2}$$
where $E_{i,s}$ represents the vector of the $ith$ member of perturbed emissions for species $s$, $E_s^p$ represents the a priori emissions
for this species, $\circ$ denotes the schur product and $N_{ens}$ denotes the ensemble size. In this way, the adjustment of emissions is
equivalent to the adjustment of perturbation factors.
In terms of the uncertainty in chemical concentrations, considering that emission uncertainty is the major contributor to
the uncertainties in air quality modelling, especially during the COVID-19 period when emissions changed rapidly,
uncertainties in chemical variables were obtained through ensemble simulations driven by perturbed emissions. The ensemble
size was chosen as 50 to maintain the balance between the filter performance and computational cost. After the ensemble
simulations, emissions of multiple species were updated using a deterministic form of EnKF (DEnKF) proposed by Sakov and
Oke (2008), which is formulated by
$$\overline{x^a} = \overline{x^b} + P_e^b H^T \left( H P_e^b H^T + R \right)^{-1} \left( y^o - H \overline{x^b} \right) \tag{3}$$
$\overline{x^b} = \frac{1}{N}\sum_{i=1}^{N} x_i^b \; ; X_i^b = x_i^b - \overline{x^b}$ (4)
$\mathbf{P_e^b} = \frac{1}{N-1}\sum_{i=1}^{N} X_i^b(X_i^b)^{\mathrm{T}}$ (5)
where $x$ denotes the state variables; $b$ the background state (a priori); $a$ the analysis state (posteriori); $\mathbf{P_e^b}$ the ensemble-
estimated background error covariance matrix and $N$ the ensemble size. $y^o$ represents the vector of observations with an error
covariance matrix of $\mathbf{R}$. $\mathbf{H}$ is the linear observational operator that maps the m-dimensional state vector $x$ to a p- (number of
observations) dimensional observational vector ($\mathbf{H}\overline{x^b}$). The state variables were defined as follows according to state
augmentation method during the assimilation:
$x_i = [c_i, \; \beta_i]^T, i = 1,2, \cdots N_{ens}$ (6)
$c_i = [PM_{2.5}, PM_{10-2.5}, \; NO_2, \; SO_2, CO]_i$ (7)
$\beta_i = [\beta_{PMF}, \beta_{BC}, \; \beta_{OC}, \; \beta_{PMC}, \beta_{NO_x}, \beta_{SO_2}, \beta_{CO}]_i$ (8)
where $x_i$ represents the $ith$ member of the assimilated state variable, which consists of the fields of chemical variables $c_i$ and
emission perturbation factors $\beta_i$. Detailed descriptions of the model state variables are summarized in Table 1. The use of
$PM_{10-2.5}$ ($PM_{10}$ minus $PM_{2.5}$) values was aimed to avoid the potential cross-correlations between $PM_{2.5}$ and $PM_{10}$ (Peng et al.,
2018; Ma et al., 2019). Moreover, to prevent spurious correlations between non- or weakly related variables, similar to Ma et
al. (2019) and Miyazaki et al. (2012), state variable localization was used during assimilation, with observations of one
particular species only used in the updates of the same species' emission rate. Corresponding relationship between the chemical
observations and adjusted emissions is summarized in Table 1. The $PM_{2.5}$ observations were one exception and were used to
update the emissions of PMF (fine mode unspeciated aerosol), BC (black carbon) and OC (organic carbon) since the
observations of speciated $PM_{2.5}$ were not available in this study. The lack of speciated $PM_{2.5}$ observations may lead to
uncertainties in the estimated emissions of PMF, BC and OC. Therefore, we only analyzed the emissions of $PM_{2.5,}$ which were
the sum of the emissions of these three species. Similarly, only $PM_{10}$ emissions were analyzed in this study, which includes
the emissions of $PM_{2.5}$ and PMC (coarse mode unspeciated aerosol).
Due to the strict control measures implemented during the last decades, the emissions in China decreased dramatically
from 2010 to 2020, especially for $SO_2$. Thus, there are large biases in the a priori estimates of emissions in China (Zheng et
al., 2018), which would lead to incomplete adjustments of the a priori emissions and degrade the performance of assimilation.
Therefore, an iteration inversion scheme was developed in this study to address the large biases of $SO_2$ emissions. As illustrated
in Fig. 2, the main idea of the iteration inversion scheme is to update the ensemble mean of the state variable using the inversion
results of the $kth$ iteration and corresponding simulations. The state variable used in the $(k + 1)th$ inversions is written as
follows:
$x_i^{k+1} = \left[c^k + c_i^e - \overline{c^e}, \beta^k + \beta_i^e - \overline{\beta^e}\right]^T$ (9)

where $c^k$ represents the simulation results using the inversed emissions of the $kth$ iteration, $c_i^e$ represents the $ith$ member of ensemble simulations with an ensemble mean of $\bar{c}^e$, $\beta^k$ represents the perturbation factors of the $kth$ iteration, and $\beta_i^e$ represents the $ith$ member of the ensemble of perturbation factors with a mean value of $\overline{\beta^e}$.

Using this method, the problems of large biases in the a priori emissions were well addressed as exemplified in Fig. 3 for SO$_2$ emissions. It can be clearly seen that due to the large positive biases in the a priori SO$_2$ emissions, the model still has large positive biases (NMB = 30.9–220.5%) and errors (RMSE = 8.7–23.0 $\mu g/m^3$) in simulated SO$_2$ concentration over all regions of China even after assimilation (first iteration). However, the biases and errors continued to decrease with the increasing of iteration times till the fourth iteration in which there were no significant improvement in SO$_2$ simulations compared to those in third iteration. These results suggested that the iteration inversion method used in this study can well constrain the a priori emission with large biases and, in this application, conducting three iteration is enough for constraining the emission. Besides SO$_2$ emissions, the iteration inversion scheme was also applied to the emissions of other species. Meanwhile, to reduce the influences of random model errors (e.g., errors in meteorological inputs) on the estimation of the variation in emissions, a 15-day running average was performed on our daily inversion results after the inversion estimation.

## 2.4 Quantification of the effects of emission changes and meteorological variations

In previous studies, the meteorological-induced (MI) changes were usually determined by the CTM with a fixed emission input setting and a varying meteorological input. Then, the difference between the MI changes and total changes in air pollutant concentrations is defined as emission-induced (EI) changes. Another approach to estimate EI changes is to perform simulations with a fixed meteorological input setting and varying emission inputs. Then, the MI changes are defined as the difference between EI changes and total changes in air pollutant concentrations. Due to the nonlinear effects of atmospheric chemical systems, these two methods yield different results. Thus, both methods were used in this study to account for the nonlinear effects. The averaged results of these two methods are used to represent the impacts of emission changes and meteorological variation on the air quality changes during the COVID-19 restrictions. In total, three scenario experiments were designed based on our inversion results (Table 2). The first scenario simulation used the varying meteorological and emission inputs from the P1 to P2 period, which represents the real-world scenario and is used to estimate the total changes in air pollutant concentrations induced by emissions and meteorological changes from the P1 to P2 period (BASE scenario). The second scenario experiment used the varying meteorological inputs but replaced the emissions during the P2 period with those during the P1 period, which was used to estimate the MI changes using the first method (MET change scenario). The third scenario experiment used the varying emissions input and replaced the meteorological input during the P2 period with that during the P1 period, which was used to estimate the EI changes using the second method (EMIS change scenario). Based on the first method, the MI and EI changes can be estimated as follows:

$$MI_{MET\ change\ scenario} = conc_{p2,MET\ change\ scenario} - conc_{p1,MET\ change\ scenario} \tag{10}$$

$$EI_{MET\ change\ scenario} = conc_{p2,BASE\ scenario} - conc_{p1,BASE\ scenario} - MI_{MET\ change\ scenario} \tag{11}$$

where $MI_{MET\ change\ scenario}$ represents the MI changes estimated based on the results from the MET change scenario,
$conc_{p1,MET\ change\ scenario}$ and $conc_{p2,MET\ change\ scenario}$ represent the averaged concentrations of air pollutants during the P1
and P2 periods under the MET change scenario, $EI_{MET\ change\ scenario}$ represents the EI changes estimated based on the results
from the MET change scenario, and $conc_{p1,BASE\ scenario}$ , $conc_{p2,BASE\ scenario}$ respectively represent the averaged
concentrations of air pollutants during the P1 and P2 periods under the BASE scenario. Similarly, the MI and EI changes
estimated based on the second method are formulated as follows:
$EI_{EMIS\ change\ scenario} = conc_{p2,EMIS\ change\ scenario} - conc_{p1,EMIS\ change\ scenario}$           (12)
$MI_{EMIS\ change\ scenario} = conc_{p2,BASE\ scenario} - conc_{p1,BASE\ scenario} - EI_{EMIS\ change\ scenario}$    (13)
Then, the estimations from these two methods are averaged to estimate the contributions of meteorological change and
emission change to the changes in PM$_{2.5}$ and O$_3$ concentrations during the COVID-19 lockdown:
$MI = (MI_{EMIS\ change\ scenario} + MI_{MET\ change\ scenario})/2$           (14)
$EI = (EI_{EMIS\ change\ scenario} + EI_{MET\ change\ scenario})/2$           (15)
$contri_{met} = \frac{MI}{MI+EI} \times 100$           (16)
$contri_{emis} = \frac{EI}{MI+EI} \times 100$           (17)
where $contri_{met}$ and $contri_{emis}$ represent the relative contributions (%) of the meteorological variations and emission
changes to the changes in air pollutant concentrations. Detailed definition of each notation used in the calculation of MI and
EI is given in Table 3.
**3 Results**
**3.1 Validation of the inversion results**
We firstly validate our inversion system by using a cross-validation method, in which 20% of observation sites were
withheld from the emission inversion and used as the validation datasets. Figure S1–6 showed the concentrations of different
air pollutants in China from 1$^{st}$ Jan to 29$^{th}$ Feb 2020 obtained from observations at validation sites and simulations using a
priori and a posteriori emission. Commonly used statistical evaluation indices, including correlation coefficient (R), mean bias
error (MBE), normalized mean bias (NMB) and root of mean square error (RMSE) are summarized in Table S1. The validation
results suggest that the posteriori simulation agreed well with the observed concentrations for all species. The large biases in
the a priori simulation of PM$_{2.5}$, PM$_{10}$, SO$_2$ and CO were almost completely removed in the a posteriori simulation with NMB
about -3.9–15.7% for PM$_{2.5}$, -3.1–11.6% for PM$_{10}$, -12.6–5.3% for NO$_2$, -9.5–6.2% for SO$_2$ and -10–7.6% for CO (Table S1).
RMSE values were also significantly reduced in the a posteriori simulation which were 9.1–32.2μg/m$^3$ for PM$_{2.5}$, 12.6–
42.4μg/m$^3$ for PM$_{10}$, 5.1–12.3μg/m$^3$ for NO$_2$, 1.2–5.6μg/m$^3$ for SO$_2$ and 0.10–0.46mg/m$^3$ for CO. Moreover, the inversion
emission considerably improved the fit to the observed time evolution of air pollutants' concentrations. The R values were

improved for all species in the a posteriori simulation that were up to 0.74–0.94 for $PM_{2.5}$, 0.63 – 0.92 for $PM_{10}$, 0.76–0.94 for $NO_2$, 0.23–0.79 for $SO_2$ and 0.63–0.92 for CO. These results suggest that our inversion results have excellent performance in representing the magnitude and variation of these species' emission in China during COVID-19 restrictions. Model performance in simulating $O_3$ concentration is relatively poor compared to other species although improvement was remarkable in NCP, NE and SE regions. This would be due to the use of outdated emission inventory for base year 2010 and that the emission of non-mental volatile organic compounds (NMVOC), another important precursor for $O_3$, were not constrained in this study. As shown in fig. S7, the NMVOC emissions for base year 2010 were generally lower than those for 2018 except over the SW regions. Considering the increasing trend of NMVOC emissions in China (Li et al., 2019), the underestimates of NMVOC emissions for base year 2020 could be larger. This is in line with the negative biases in the simulated $O_3$ concentrations over these regions.

## 3.2 Emission changes of multi-species during COVID-19 restrictions

### 3.2.1 Unbalanced emission changes between $NO_x$ and other species

The control of COVID-19 began on 23rd January when the Chinese government declared the first level of national responses to public health emergencies, one day before the 2020 Chinese New Year Eve. Figure 4 shows the time evolution of the normalized emission anomaly for different species in China from 1st January to 29th February. The temporal variation in the emission varied largely between $NO_x$ and other species. Due to the combined effects of the Spring Festival and COVID-19 lockdown, $NO_x$ emissions decreased continuously at the beginning of January until approximately one week after the implementation of the COVID-19 lockdown, with estimated decreases in $NO_x$ emissions of up to 42.5% from the P1 to P2 period (Table 4). Subsequently, the $NO_x$ emissions stabilized with small fluctuations until the official back-to-work day when the $NO_x$ emissions began to increase due to easing of the control measures and the resumption of business. According to inversion estimation, $NO_x$ emissions recovered by 3.9% during the P3 period. These results indicate that the temporal variation in our estimated $NO_x$ emissions agreed well with the timing of the Spring Festival and different control stages of COVID-19. However, for other species (i.e., $PM_{2.5}$, $PM_{10}$, $SO_2$ and CO), although their emissions generally decreased from 1st January to the end of the 2020 Spring Festival holiday, they showed much smaller reductions than the $NO_x$ emissions. The emission reduction for these species was only approximately 7.9-12.1% (Table 4). This is consistent with the inversion results by Hu et al. (2022) who found that $SO_2$ emissions in China decreased only by 9.2% during COVID-19 lockdown. In addition, the emissions of these species quickly rebounded to their normal level just one week after the end of the Spring Festival holiday. As estimated by our inversion results, the $SO_2$ emissions recovered by 7.2% during the P3 period, which was only 2.5% lower than that during the P1 period. The $PM_{2.5}$ and $PM_{10}$ emissions during the P3 period were 3.3% and 43.6% higher, respectively, than those during the P1 period.

Similar results were found in different regions of China (Fig. 5 and Table 5), where the $NO_x$ emissions decreased much more than other species. In addition, unlike the uniform decreases in $NO_x$ emissions in different regions of China (~40%),

there was apparent spatial heterogeneity in the emission changes in $PM_{2.5}$, $PM_{10}$, $SO_2$ and CO (Table 5 and Fig. 6). For example, from the P1 to P2 period, the $PM_{2.5}$ emissions decreased by over 20% in the Central region but only by 8.8% in the NE region. The $PM_{2.5}$ emissions even increased by 5.5% in the NCP region. This may be due to the increased emissions from industry and fireworks according to the field measurements conducted by previous studies (Li et al., 2022; Ma et al., 2022; Zuo et al., 2022; Dai et al., 2020). Based on the measurement of stable Cu and Si isotopic signature and distinctive metal ratios in Beijing and Hebei, Zuo et al. (2022) analyzed the variations in the $PM_{2.5}$ sources during the COVID-19 pandemic, who reported that the primary $PM_{2.5}$ emissions did not decrease in Beijing and Hebei, and that the PM-associated industrial emissions may actually increase during the lockdown period. The increased industrial heat sources detected by Li et al. (2022) based on VIIRS active fire data also supported the increased industrial emissions over the NCP region during lockdown period. Meanwhile, consistent with the field measurements in Beijing and Tianjin conducted by Ma et al. (2022) and Dai et al. (2020), substantial high levels of potassium ($K^+$) and magnesium ($Mg^{2+}$) ion were found over the NCP region during the Spring Festival according to the aerosol chemical composition measurements obtained from CNEMC (Fig. S8). Since $K^+$ and $Mg^{2+}$ are two important fingerprints of the firework emissions, the high levels of $K^+$ and $Mg^{2+}$ suggest that the emissions from fireworks during Spring Festival were also a potential contributor to the increased of $PM_{2.5}$ emissions over the NCP region. In contrast, the SW and central regions exhibited relatively larger emission reductions for these species (Fig. 5 and Table 5) by 12.6–25.9% and 10.6–23.7%, respectively. The emission rebound during the P3 period was more prominent in the SE, central and SW regions (Fig. 5 and Fig. 7), where emissions recovered by 6.0–16.4% for $NO_x$, 7.5–19.8% for $SO_2$, 7.4–13.1% for CO, 12.3–47.7% for $PM_{2.5}$ and 28.6–135.9% for $PM_{10}$ (Table 5). This result is consistent with the earlier degradation of the response level to the COVID-19 virus (from the first level to the second or third level) over these regions (Table S2). In contrast, there were decreases in emissions in the NCP, NE and NW regions. $PM_{2.5}$ emissions were reduced by 9.9% in the NCP region and by 19.2% in the NE region from the P2 to P3 period (Table 5). Moreover, we found that the $PM_{10}$ emissions surged in the NW and central regions, where the $PM_{10}$ emissions during the P3 period were almost two times larger than those during the P2 period (Table 5). However, this finding may be related to the enhanced dust emissions over these two regions rather than the effects of returning to work according to the decreased $PM_{2.5}/PM_{10}$ ratios during the P3 period. According to Fig.S9, the $PM_{2.5}/PM_{10}$ ratio was relatively stable during the P1 and P2 period, but it decreased substantially during the P3 period, from 0.81 to 0.48 over the NW region and from 0.77 to 0.53 over the Central region. A lower $PM_{2.5}/PM_{10}$ ratio commonly suggests that the $PM_{10}$ is more likely to be attributed to natural sources such as dust (Wang et al., 2015; Fan et al., 2021). Moreover, the NW and Central region are typical source areas of dust in China, therefore the increasing of $PM_{10}$ emissions over NW and Central regions may be mainly related to the enhanced dust emissions. This demonstrates the necessity to consider changes in natural emissions during COVID-19 restrictions. Thus, to reduce the effects of natural emissions on our findings, the same analysis was performed for the emissions over southeast China (Fig. S10) where emissions were dominated by anthropogenic sources, which shows consistent results with the findings above (Fig. S11 and Table S3).

### 3.2.3 Explanations for the emission changes during COVID-19 restrictions

Two explanations may help clarify the unbalanced emission changes between $NO_x$ and other species. First, the COVID-19 lockdown policy has led to dramatic decreases in transportation activities throughout China; however, as shown in Fig. 4, the relative contributions of the transportation sector to the emissions of $SO_2$ (2.4%), CO (18.5%), $PM_{2.5}$ (6.1%) and $PM_{10}$ (4.7%) are much smaller than those for $NO_x$ emissions (34.3%) (Zheng et al., 2018; Li et al., 2017a). Thus, the reduction in traffic activities can only substantially decrease $NO_x$ emissions. Reductions in CO emissions (-10.6%) were relatively larger than those for $SO_2$ (-9.7%) and $PM_{2.5}$ (-7.9%) emissions, which is consistent with the relatively larger contributions of the transportation sector to CO emissions. However, the differences in the percentage decreases in emissions of CO, $SO_2$ and $PM_{2.5}$ is not as significant as the differences in their transportation share (18% versus 2% and 6%). This may be on the one hand due to the uncertainty in the estimated relative contributions of different sectors to the total emissions of CO, $SO_2$ and $PM_{2.5}$, on the other hand were possibly due to the uncertainty in the emission inversions, especially considering that the decreasing trend of CO, $SO_2$ and $PM_{2.5}$ were not significant. Also, other factors beyond transportation may have influenced the reductions of anthropogenic emissions during P2 period. For example, the $PM_{10}$ emissions showed the largest reductions among these four species, which is related in part to the reduced dust emissions due to shutting down of construction sites during the lockdown period (Li et al., 2020). Second, as shown in Fig. 4, the industrial and residential sectors are the major contributors to the anthropogenic emissions of $SO_2$, CO, $PM_{2.5}$ and $PM_{10}$ in China, together contributing 77.6%, 78.3%, 86.5% and 86.3%, respectively, to their total emissions. The much smaller reductions of these species' emissions were thus in line with the fact that there were no intentional restrictions on heavy industry during the COVID-19 restrictions. A large number of non-interruptible processes, such as steel, glass, coke, refractory, petrochemical, electric power, and especially heating, cannot be stopped during the COVID-19 lockdown. According to statistical data from the National Bureau of Statistics of China (Fig. S12), the industrial and power sectors did not show similar reductions in their activity levels as those seen in the transportation sector. Power generation and steel production even showed increases in many provinces, which corresponds well with the emission increases over these regions. In addition, since people were required to stay at home, residential emissions were likely increased due to the increased energy consumption for heating or cooking. Therefore, our inversion results supported the views that the emissions of species related to industrial and residential activities did not decline much during the lockdown period, and that the COVID-19 lockdown policy was largely a traffic control measure with small influences on other sectors.

### 3.3 Investigation of air quality change over the NCP region during COVID-19 restrictions

Using the inversion results, we reassessed the environmental impacts of the COVID-19 restrictions on the air pollution over NCP region. The NCP region was chosen because it is the key target region of air pollution control in China and where unexpected severe haze occurred. A major caveat in previous studies that explored the impacts of COVID-19 lockdowns on air quality is the uncertainty in the emission changes during COVID-19 restrictions. The inversion results enable us give a more reliable assessment of the environmental impacts of COVID-19 restrictions. Figure 8 shows the observed changes in

PM$_{2.5}$ and O$_3$ concentrations over the NCP region from the P1 to P2 period. The observations showed consistent reductions in
PM$_{2.5}$ concentrations over the NCP region (by 13.6 μg/m$^3$). However, substantial increases in PM$_{2.5}$ concentrations were
observed in the Beijing area (by 31.2 μg/m$^3$). In contrast to the widespread reductions in PM$_{2.5}$ concentrations, the O$_3$
concentrations significantly increased over the whole NCP region (by 28.3 μg/m$^3$) and the Beijing area (by 16.8 μg/m$^3$). The
simulations based on our inversion results reproduced the observed changes in PM$_{2.5}$ and O$_3$ concentrations over the NCP
region well, although the O$_3$ concentrations were underestimated in all regions (Fig. S6) and the changes in PM$_{2.5}$ and O$_3$
concentrations were slightly overestimated by 1.6 and 2.6 μg/m$^3$ in the simulation (Fig. 8).
As detailed in the Sect 2.4, the simulated changes in air pollutant concentrations before and after lockdown were
decomposed into meteorological-induced (MI) changes and emission-induced (EI) changes through two different scenarios to
account for the nonlinearity of the atmospheric chemical system. According to Fig. S13, the differences in calculated MI and
EI based on different scenarios were small for PM$_{2.5}$ concentrations, which were about 2 μg/m$^3$ in this application, while they
were relatively larger for O$_3$, which were around 5 μg/m$^3$ over the Beijing and NCP region (Fig. S14). In addition, the sign of
calculated MI using different scenarios were opposite although both suggested weak contributions of meteorological variation
to the changes of O$_3$ concentrations. This suggests that the calculated MI and EI changes of O$_3$ concentrations could be more
sensitive to the used scenarios, which may be associated with the stronger chemical nonlinearity of the O$_3$ concentrations.
Figure 9 shows the mean results of the calculated MI and EI changes using the two different scenarios. It shows that the
meteorological variation dominated the changes in PM$_{2.5}$ concentrations over the NCP region, which contributed 90% of the
PM$_{2.5}$ reductions over most parts of the NCP region. Moreover, this variation made significant contributions (57.9%) to the
increases in PM$_{2.5}$ concentrations over the Beijing area. This finding suggested that meteorological variations played an
irreplaceable role in the occurrence of the unexpected PM$_{2.5}$ pollution around the Beijing area. Compared with the
meteorological conditions before lockdown (Fig. 10), there were increases in relative humidity over northern China, which
facilitated the reactions for aerosol formation and growth. Wind speed also decreased over the Beijing area accompanied by
an anomalous south wind, which facilitated aerosol accumulation and the transportation of air pollutants from the polluted
industrial regions of the Hebei Province to Beijing. The increases in boundary layer height from the P1 to P2 period were also
much smaller in the Beijing area than in other areas of the NCP. Thus, the Beijing area has exhibited distinct meteorological
variations from other areas of the NCP region, which correspond well to the different changes in PM$_{2.5}$ concentrations over the
Beijing area.
The emission changes contributed slightly to the PM$_{2.5}$ reductions over the NCP region (8.6%). This is because, on the
one hand, the large reductions in NO$_x$ emissions (by 44.4%) only reduced nitrate by approximately 10–30% due to the nonlinear
effects of chemical reactions (Fig. 11), and on the other hand, the emissions of primary PM$_{2.5}$ and its precursors from other
sectors changed little during the COVID-19 restrictions (Table 5). The emission changes contributed more to the increased
PM$_{2.5}$ concentrations over the Beijing area (42.1%). This is mainly associated with the increases in primary PM$_{2.5}$ emissions
around the Beijing area, as seen in Fig. 6, possibly due to the increased emissions from the industry as we mentioned before
(Zuo et al., 2022) and the increased firework emissions during the Spring Festival as shown by the rapid increases in

concentrations of $K^+$ and $Mg^{2+}$ measured by CNEMC (Fig. S15). Therefore, our results suggested that the unexpected $PM_{2.5}$ pollution during lockdown period was mainly driven by unfavorable meteorological conditions together with small changes or even increases in primary $PM_{2.5}$ emissions. This finding is in line with previous results of Le et al. (2020) but different from those of Huang et al. (2021), who suggested that enhanced secondary aerosol formation was the main driver of severe haze during the COVID-19 restrictions. To investigate it, we further analyzed the changes in the concentrations of secondary inorganic aerosols (SIAs). First, we evaluated our model results against the observed SIA concentrations, which showed that the model results using our inversion emissions well reproduced the observed concentrations of SIAs over the NCP region (Fig. 12) with mean bias (MB) ranging from -5.14 to 5.45 μg/m$^3$ and correlation coefficient (R) ranging from 0.59 to 0.80. The observed increases in SIA concentrations over the Beijing area, especially for sulfate concentrations, were also captured in our simulations (Fig. 11), although underestimation occurred due to the uncertainty in simulating SIA concentrations. Through sensitivity experiments, we found that the increases in SIA concentrations were still driven by meteorological variations (Fig. 13). In fact, the emission reductions only led to a 10% decrease in SIA concentrations over the NCP region. This finding suggests that the enhanced secondary aerosol formation was likely mainly driven by the unfavorable meteorological conditions associated with higher temperature and relative humidity instead of the emission reductions during the lockdown period. This is in line with the observation evidences from Ma, T et al (2022) who emphasized that the increased temperature and relative humidity promoted the formation of secondary pollutants during the COVID-19 restrictions.

In terms of $O_3$ concentrations, the emission changes subsequently became the dominant contributor to the $O_3$ increases by more than 100% in the Beijing area and by 96.0% over the NCP region. This result is mainly because the lockdown period occurred in midwinter when photochemical $O_3$ formation was minimal; thus, the large increase in $O_3$ is expected solely from the effect of the reduced titration reaction associated with the large reductions in $NO_x$ emissions. Although the higher temperature and slower wind speed during the lockdown period were favorable for the increases in $O_3$ concentrations, their contributions were much smaller than those of emission changes (Fig. 9). These results suggested that control measures, such as COVID-19 restrictions, were inefficient for air pollution mitigation in China considering the high economic cost of the COVID-19 restrictions.

We also compared our results with previous studies that differentiated the contributions of meteorology and emission to the $PM_{2.5}$ and $O_3$ concentrations. Before comparisons, it should be noted that it is difficult to directly compare our results with previous studies due the altered definition of meteorological contribution, different reference period that used to quantify the meteorological contributions and different targeted region. For example, in Song et al. (2021), the reference period used to determine the meteorological contribution is the corresponding period of COVID-19 pandemic in 2019. Le et al. (2020) used the multiyear climatology as the reference period. In Wang et al. (2020) and Sulaymon et al. (2021), the MI changes of $PM_{2.5}$ concentrations were defined as the difference between the modeled concentrations in high-pollution days and those in low-pollution days under hypothetical emission reduction scenario. Zhao et al. (2020) used a similar reference period to ours to determine the MI changes but they used the outdated emission inventory. Table 6 summarized the results from the selected studies over Beijing and Beijing-Tianjin-Hebei region. Note that some studies only provided the relative changes in the

modeled PM$_{2.5}$ concentrations. It shows that due to the uncertainties in emission changes during COVID-19 pandemic, the EI
changes estimated by Zhao et al. (2020) were possibly overestimated compared to our studies (55% versus 24.7%). Both
Sulaymon et al. (2021) and Wang et al. (2020) suggested negative EI changes during COVID-19 period in Beijing. This
because they presumed that the emissions were largely reduced during COVID-19 lockdown which may deviate from the real
changes of emissions according to our inversion results. Meanwhile, although they used same method and reference period,
their results differed largely (-2.7 versus -13.4 $\mu g/m^3$) due to the different emission reduction scenario they assumed. Le et
al. (2020) only considered the emission reductions of NO$_x$ in their sensitivity simulations without considerations of other
species, therefore their calculated EI changes may be underestimated compared to our results (almost 0% versus 24.7%).
However, the calculated MI changes were consistent between our study and Le et al. (2020). In terms of O$_3$, the calculated EI
changes by our study were also higher than that calculated by Zhao et al. (2020) in Beijing (85.7% versus 70%). These results
suggested that the EI and MI changes calculated by our study could be more reasonable, considering that the emissions of
different species were well constrained which could better represent the temporal variation and spatial heterogeneity of
emission changes during COVID-19.
**4 Conclusions and discussions**
The COVID-19 pandemic is an unprecedented event that significantly influenced the social activity and associated
emissions of air pollutants. Our results provide a quantitative assessment of the influences of COVID-19 restrictions on multi-
air pollutant emissions in China. Otherwise, understanding of the relationship between air quality and human activities may
be biased. The inversion results provide important evidences that the COVID-19 lockdown policy was largely a traffic control
measure with substantially reducing impacts on NO$_x$ emissions but much smaller influences on the emissions of other species
and other sectors. Traffic control has widely been considered to be the normal protocol in implementing regulations in many
cities of China, but its effectiveness on air pollution control is still disputed (Han and Naeher, 2006; Zhang et al., 2007; Chen
et al., 2021; Cai and Xie, 2011; Chowdhury et al., 2017; Li et al., 2017c). Thus, the COVID-19 restrictions provided us with a
real nationwide traffic control scenario to investigate the effectiveness of traffic control on the mitigation of air pollution in
China. The results suggested that traffic control as a separate pollution control measure has limited effects on the coordinated
control of high concentrations of O$_3$ and PM$_{2.5}$ under the current air pollution conditions in China. In this case, the PM$_{2.5}$
concentrations were slightly reduced, while leading to substantial increases in O$_3$ concentrations. Severe haze was also not
avoided during the COVID-19 restrictions due to unbalanced emission changes from other sectors and unfavorable
meteorological conditions. China is now facing major challenges in both controlling PM$_{2.5}$ and controlling emerging O$_3$
pollution. The tragic COVID-19 pandemic has revealed the limitation of the road traffic control measure in the coordinated
control of PM$_{2.5}$ and O$_3$. More comprehensive regulations for multiple precursors from different sectors are required in the
future to address O$_3$ and PM$_{2.5}$ pollution in China.
Finally, there are certain limitations that should be aware of in our inversion work. Firstly, the COVID-19 restrictions
were initiated during the Spring Festival of China which would also influence the air pollutant emissions in China. However,
the inversion method used in this study did not differentiate the contributions of the Spring Festival from the COVID-19
restrictions. Similarly, the effects of natural emission changes were not differentiated in this study, which would lead to
uncertainty in quantifying the effects of the COVID-19 restrictions on air pollutant emissions. Secondly, the overestimations
of $NO_2$ measurement induced by chemiluminescence monitor interference were not directly corrected in our study due to the
lack of synchronous observations of $HNO_3$, PAN and AN, thus the estimated $NO_x$ emissions could be slightly overestimated
according to the sensitivity run with corrected $NO_2$ measurement using CFs (Fig. S16–18). Meanwhile, the sensitivity results
suggest that the inversed $NO_x$ emissions may even drop faster if the $NO_2$ measurement were corrected over the SE and SW
regions (Fig. S19). Thirdly, the use of outdated emission inventory as the a priori emission would also be a potential limitation
in our work although the iteration inversion method was used. A sensitivity inversion run was thus conducted based on the a
priori emission for a more recent year of 2018 to test the influence of the a priori emission inventory. This new emission
inventory is comprised of the anthropogenic emissions obtained from HTAPv3 (Crippa et al., 2023), the biogenic, soil and
oceanic emissions obtained from the CAMS global emission inventory
([https://ads.atmosphere.copernicus.eu/cdsapp#!/dataset/cams-global-emission-inventories?tab=overview](https://ads.atmosphere.copernicus.eu/cdsapp#!/dataset/cams-global-emission-inventories?tab=overview), last access: March
15, 2023) and the biomass burning emissions obtained from the Global Fire Assimilation System (GFAS) (Kaiser et al., 2012).
Detailed steps of the new inversion estimation were same as those elucidated in Sect.2. The results suggest that the inversion
results based on the 2010 and 2018 inventory were broadly close to each other, while the inversion results based on 2018
inventory were relatively higher than those based on 2010 inventory, reflecting the uncertainty in our inversion results caused
by the choice of a priori emission inventory (Fig. S20–22). However, the sensitivity run consistently showed that the $NO_x$
emissions decreased much larger than other species (Fig. S23–24). This suggests that the choice of a priori emission inventory
may not obviously influence the main conclusion of our study, but can lead to uncertainty in the magnitude of the inversion
results which should be aware of by potential readers.











**Tables**
**Table 1. Corresponding relationship between the chemical observations and adjusted emissions**

| Species | Descriptions | Observations that used for inversions of this species |
|---|---|---|
| BC | black carbon | $PM_{2.5}$ |
| OC | organic carbon | $PM_{2.5}$ |
| PMF | fine mode unspeciated aerosol | $PM_{2.5}$ |
| PMC | coarse mode unspeciated aerosol | $PM_{10} - PM_{2.5}$ |
| $NO_x$ | nitrogen oxide | $NO_2$ |
| $SO_2$ | sulfur dioxide | $SO_2$ |
| CO | carbon monoxide | CO |


**Table 2. Configuration of simulation scenarios**

| Scenarios | Meteorology input | Emission input | Purpose |
|---|---|---|---|
| BASE scenario | varied meteorological condition from pre lockdown to lockdown period | varied emission from pre-lockdown to lockdown period | To estimate the total changes of air pollutant concentrations induced by emission and meteorological change |
| MET change scenario | varied meteorological condition from pre-lockdown to lockdown period | constant emissions during pre-lockdown and lockdown period | To estimate the impacts of meteorological changes on the air pollutants |
| EMIS change scenario | constant meteorological during pre-lockdown and lockdown period | varied emission from pre-lockdown to lockdown period | To estimate the impacts of emission changes on the air pollutants |



**Table 3. Descriptions of different items used in the calculation of meteorological-induced and emission-induced changes of air pollutant concentrations**

| notation | description |
|---|---|
| $MI$ | meteorological-induced changes in air pollutant concentrations |
| $EI$ | emission-induced changes in air pollutant concentrations |
| $MI_{MET\ change\ scenario}$ | meteorological-induced changes in air pollutant concentrations calculated by the MET change scenario |
| $EI_{MET\ change\ scenario}$ | emission-induced changes in air pollutant concentrations calculated by total changes minus $MI_{MET\ change\ scenario}$ |
| $EI_{EMIS\ change\ scenario}$ | emission-induced changes in air pollutant concentrations calculated by the EMIS change scenario |
| $MI_{EMIS\ change\ scenario}$ | meteorological-induced changes in air pollutant concentrations calculated by total changes minus $EI_{EMIS\ change\ scenario}$ |
| $conc_{p1,BASE\ scenario}$ | averaged concentrations of air pollutants during P1 period under the BASE scenario |
| $conc_{p2,BASE\ scenario}$ | averaged concentrations of air pollutants during P2 period under the BASE scenario |
| $conc_{p1,MET\ change\ scenario}$ | averaged concentrations of air pollutants during P1 period under the MET change scenario |
| $conc_{p2,MET\ change\ scenario}$ | averaged concentrations of air pollutants during P2 period under the MET change scenario |
| $conc_{p1,EMIS\ change\ scenario}$ | averaged concentrations of air pollutants during P1 period under the EMIS change scenario |
| $conc_{p2,EMIS\ change\ scenario}$ | averaged concentrations of air pollutants during P2 period under the EMIS change scenario |
| $contri_{met}$ | relative contributions of the meteorological variations to the changes in air pollutant concentrations |
| $contri_{emis}$ | relative contributions of the emission changes to the changes in air pollutant concentrations |


**Table 4. Inversion estimated emissions of different air pollutants in China and their changes between different periods during COVID-19.**

|  | $NO_x$ | $SO_2$ | CO | $PM_{2.5}$ | $PM_{10}$ |
|---|---|---|---|---|---|
| P1 (Gg/day) | 72.9 | 23.8 | 1160.2 | 44.5 | 75.5 |
| P2 (Gg/day) | 41.9 | 21.5 | 1037.4 | 40.9 | 66.4 |
| P3 (Gg/day) | 44.8 | 23.2 | 1078.2 | 45.9 | 108.4 |
| (P2-P1)/P1 | -42.5% | -9.7% | -10.6% | -7.9% | -12.1% |
| (P3-P2)/P1 | 3.9% | 7.2% | 3.6% | 11.2% | 55.7% |
| (P3-P1)/P1 | -38.6% | -2.5% | -7.0% | 3.3% | 43.6% |

**Table 5. Inversion estimated emission changes of different air pollutants over different regions in China between different periods**
**during COVID-19 restrictions**

|  | $NO_x$ | $PM_{2.5}$ | $PM_{10}$ | $SO_2$ | CO |
|---|---|---|---|---|---|
| **NCP** |  |  |  |  |  |
| (P2-P1)/P1 | -44.4% | 5.5% | 2.8% | -1.6% | -4.3% |
| (P3-P2)/P1 | -0.8% | -9.9% | 31.8% | -5.9% | -10.0% |
| (P3-P1)/P1 | -45.2% | -4.3% | 34.7% | -7.5% | -14.3% |
| **NE** |  |  |  |  |  |
| (P2-P1)/P1 | -41.8% | -8.8% | -3.5% | -3.2% | -10.9% |
| (P3-P2)/P1 | -6.0% | -19.2% | 23.7% | -2.9% | -6.6% |
| (P3-P1)/P1 | -47.8% | -28.0% | 20.2% | -6.1% | -17.5% |
| **SE** |  |  |  |  |  |
| (P2-P1)/P1 | -41.4% | -9.5% | -24.4% | -19.4% | -3.5% |
| (P3-P2)/P1 | 10.2% | 12.3% | 28.6% | 19.8% | 13.1% |
| (P3-P1)/P1 | -31.2% | 2.8% | 4.2% | 0.3% | 9.7% |
| **SW** |  |  |  |  |  |
| (P2-P1)/P1 | -43.5% | -12.6% | -25.9% | -17.5% | -23.8% |
| (P3-P2)/P1 | 6.0% | 47.7% | 33.1% | 7.5% | 7.4% |
| (P3-P1)/P1 | -37.5% | 35.1% | 7.2% | -10.0% | -16.4% |
| **NW** |  |  |  |  |  |
| (P2-P1)/P1 | -38.5% | -4.0% | -8.3% | 14.2% | -2.6% |
| (P3-P2)/P1 | -21.1% | 4.9% | 145.3% | -4.1% | -7.2% |
| (P3-P1)/P1 | -59.6% | 0.9% | 136.9% | 10.1% | -9.8% |
| **Central** |  |  |  |  |  |
| (P2-P1)/P1 | -43.8% | -23.7% | -15.7% | -10.6% | -17.4% |
| (P3-P2)/P1 | 16.4% | 24.4% | 135.9% | 18.5% | 8.4% |
| (P3-P1)/P1 | -27.4% | 0.7% | 120.3% | 7.9% | -9.0% |




**Table 6. Calculated MI and EI changes in PM$_{2.5}$ concentrations during COVID-19 pandemic by previous studies**

| | MI changes | EI changes | Region | Reference period | Method | Reference |
|---|---|---|---|---|---|---|
| 1 | 26.79 μg/m³ | -21.84 μg/m³ | Beijing | January 23-March 10, 2019 versus January 23-March 10, 2020 | observation-based wind-decomposition method | Song et al. (2021) |
| 2 | Around 20 μg/m³ | -2.7 μg/m³ | Beijing | January 01 to February 29, 2020 | CTM with hypothetical emission reduction scenario | Sulaymon et al. (2021) |
| 3 | Around 45 μg/m³ | -13.4 μg/m³ | Beijing | January 01 to February 29, 2020 | CTM with hypothetical emission reduction scenario | Wang et al. (2020) |
| 4 | 31.3% | Around 0% | Beijing-Tianjin-Hebei | January 01 to February 13, 2020 | CTM sensitivity simulations using different emission rates and multiyear climatology | Le et al. (2020) |
| 5 | Around 5% | Around 55% | Beijing | January 16-22, 2020 versus January 26 to February 1, 2020 | CTM with fixed emission inventory for 2017 | Zhao et al. (2020) |
| 6 | 17.5 μg/m³ (34.0%) | 12.7 μg/m³ (24.7%) | Beijing | January 1-20, 2020 versus January 21 to February 9, 2020 | CTM with inversion emission inventory | This study |


**Figures**

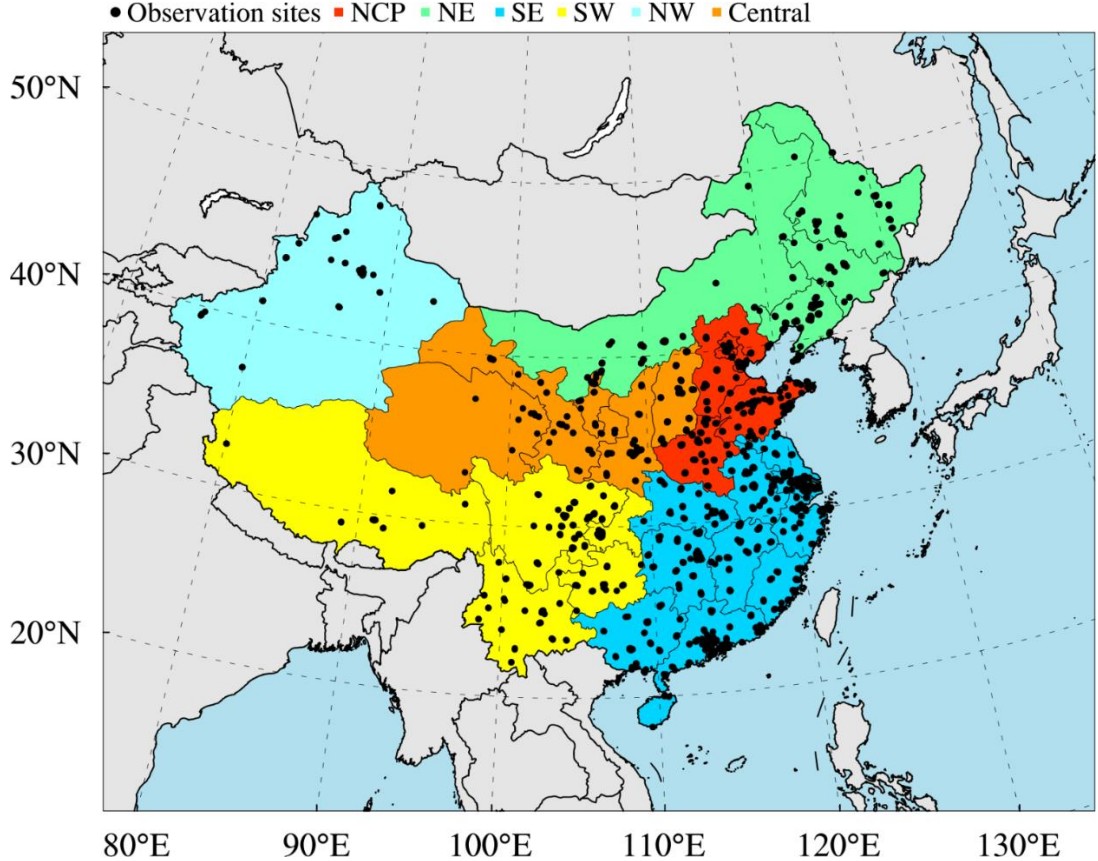


**Figure 1: Modeling domain of the ensemble simulation overlay the distributions of observation sites from CNEMC. Different colours**
**denote the different regions in mainland of China, namely North China Plain (NCP), Northeast China (NE), Southwest China (SW),**
**Southeast China (SE), Northwest China (NW) and Central.**

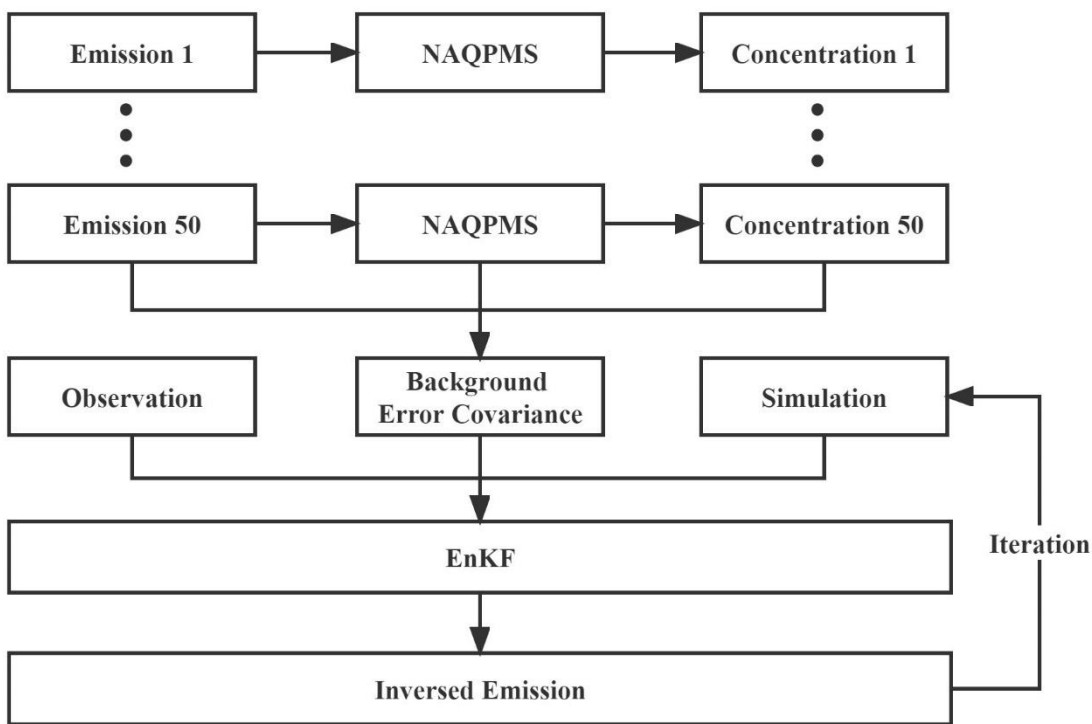


**Figure 2: Illustration of the iteration inversion scheme used in this study.**

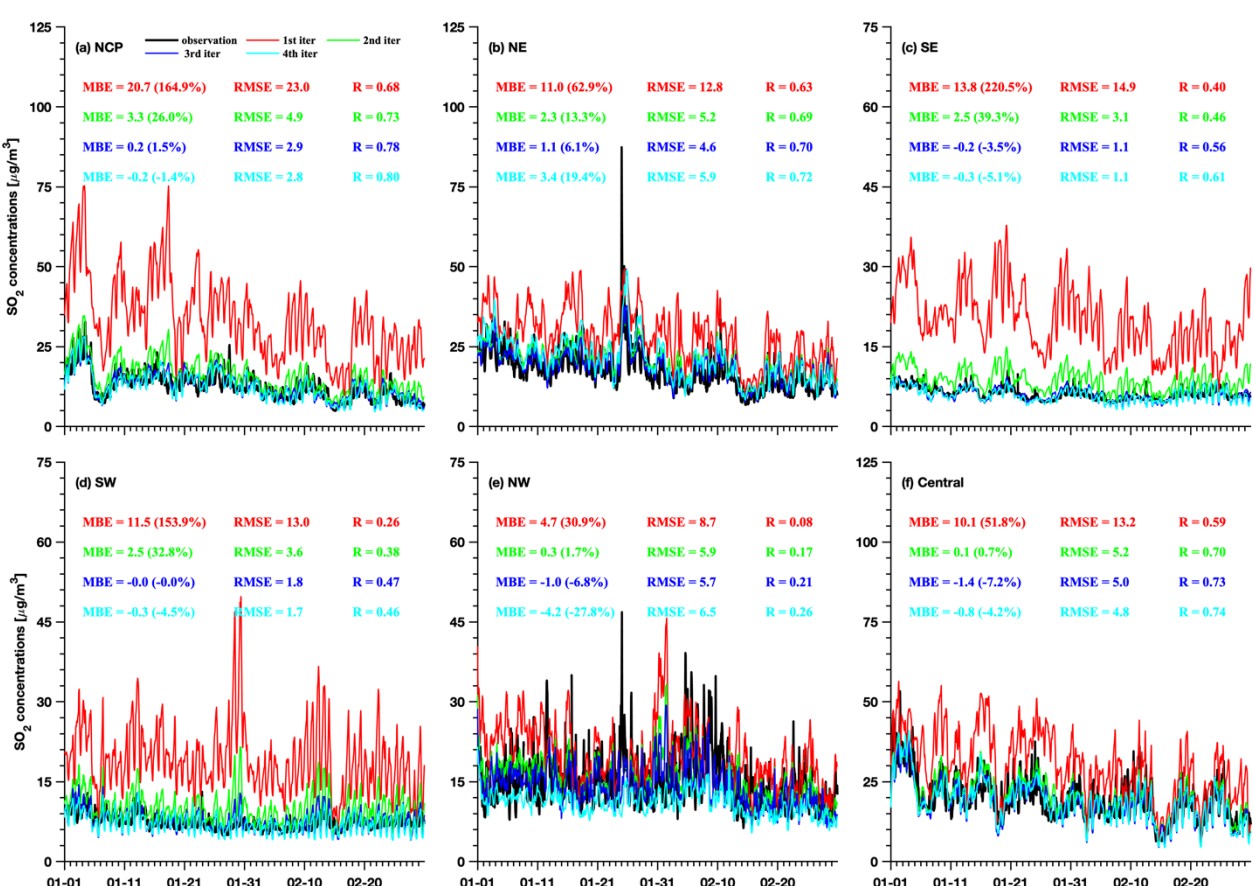

Figure 3: Comparisons of the observed and simulated mean SO₂ concentrations using emissions of different iteration time at validation sites over (a) NCP region, (b) NE region, (c) SE region, (d) SW region, (e) NW region and (f) Central region.

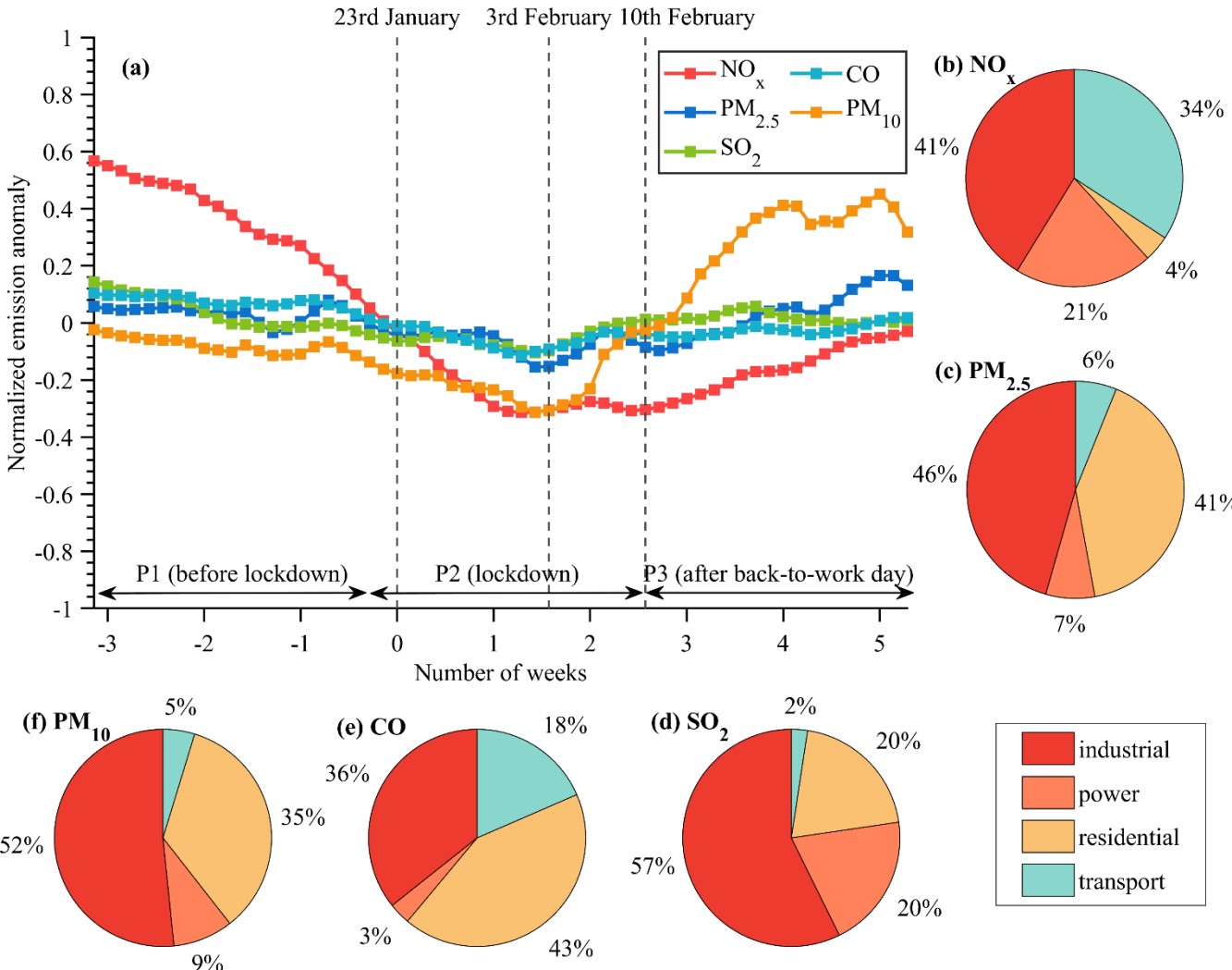

Figure 4: (a) Time series of normalized emission anomalies estimated by inversion results for different species in China from 1st
January to 29th February 2020, and (b-f) Relative contributions of different sectors to the total anthropogenic emissions of NOx,
PM2.5, PM10, CO and SO₂ obtained from Zheng et al. (2018). The normalized emission anomaly is calculated by the emission anomaly
divided by the average emissions during the whole period.

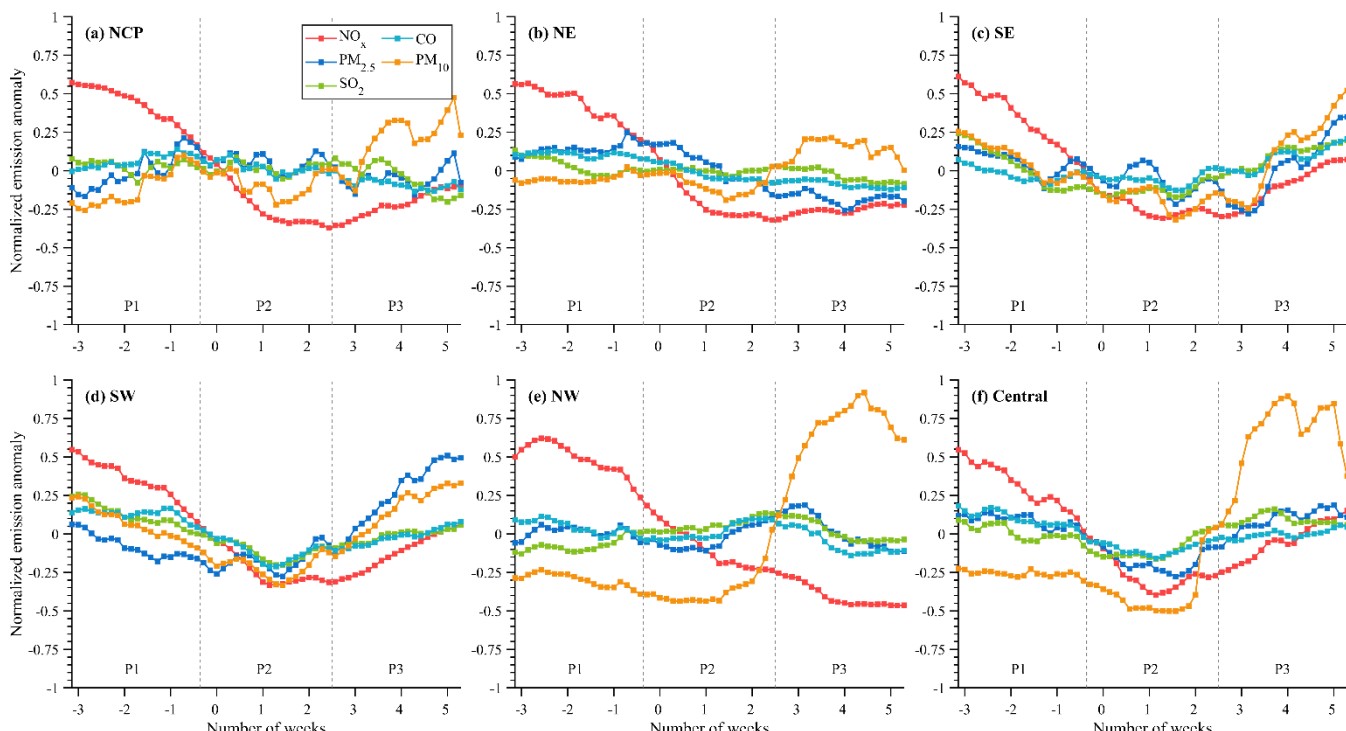

Figure 5: Time series of normalized emission anomalies estimated by inversion results for different species over (a) NCP region, (b)
NE region, (c) SE region, (d) SW region, (e) NW region and (f) Central region from 1st January to 29th February 2020.

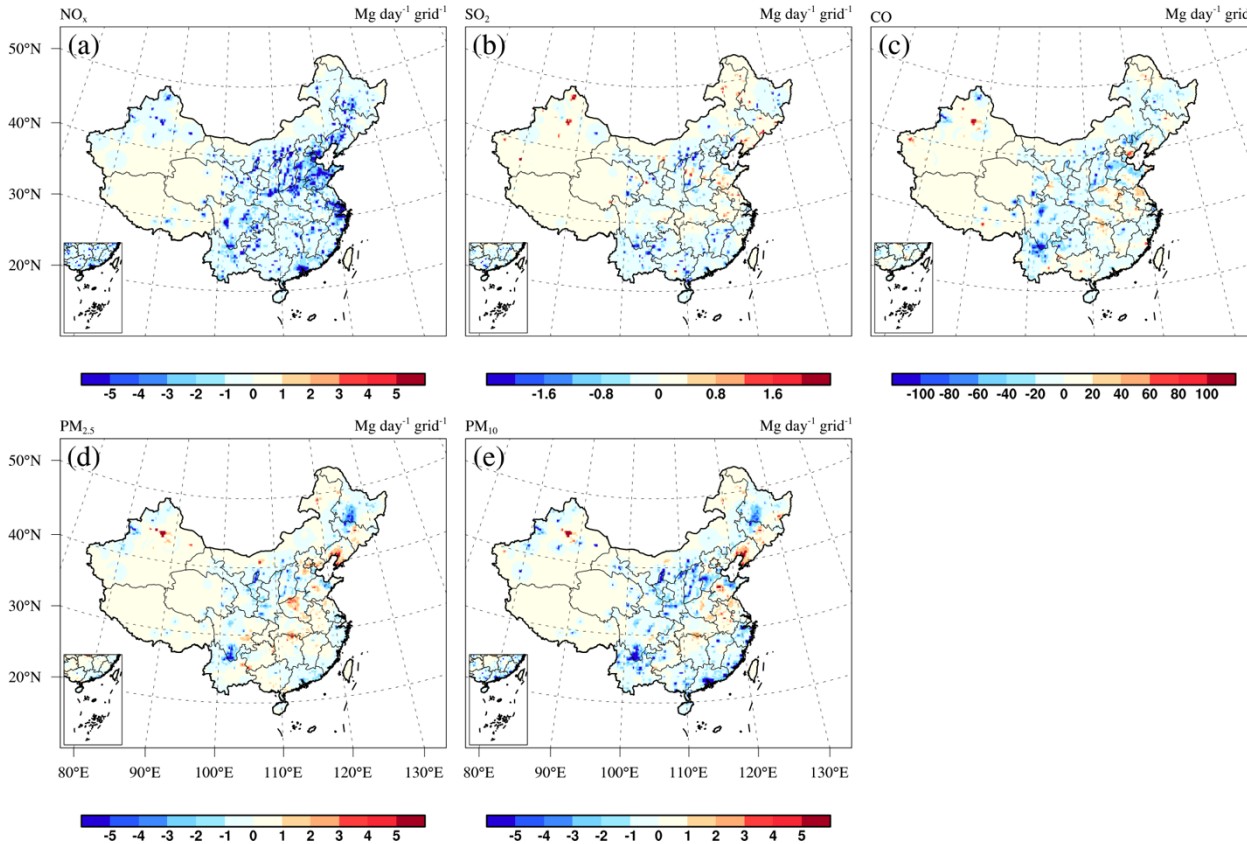


Figure 6: The inversion estimated emission changes of (a) NO$_x$, (b) SO$_2$, (c) CO, (d) PM$_{2.5}$ and (e) PM$_{10}$ in China from P1 to P2 period.

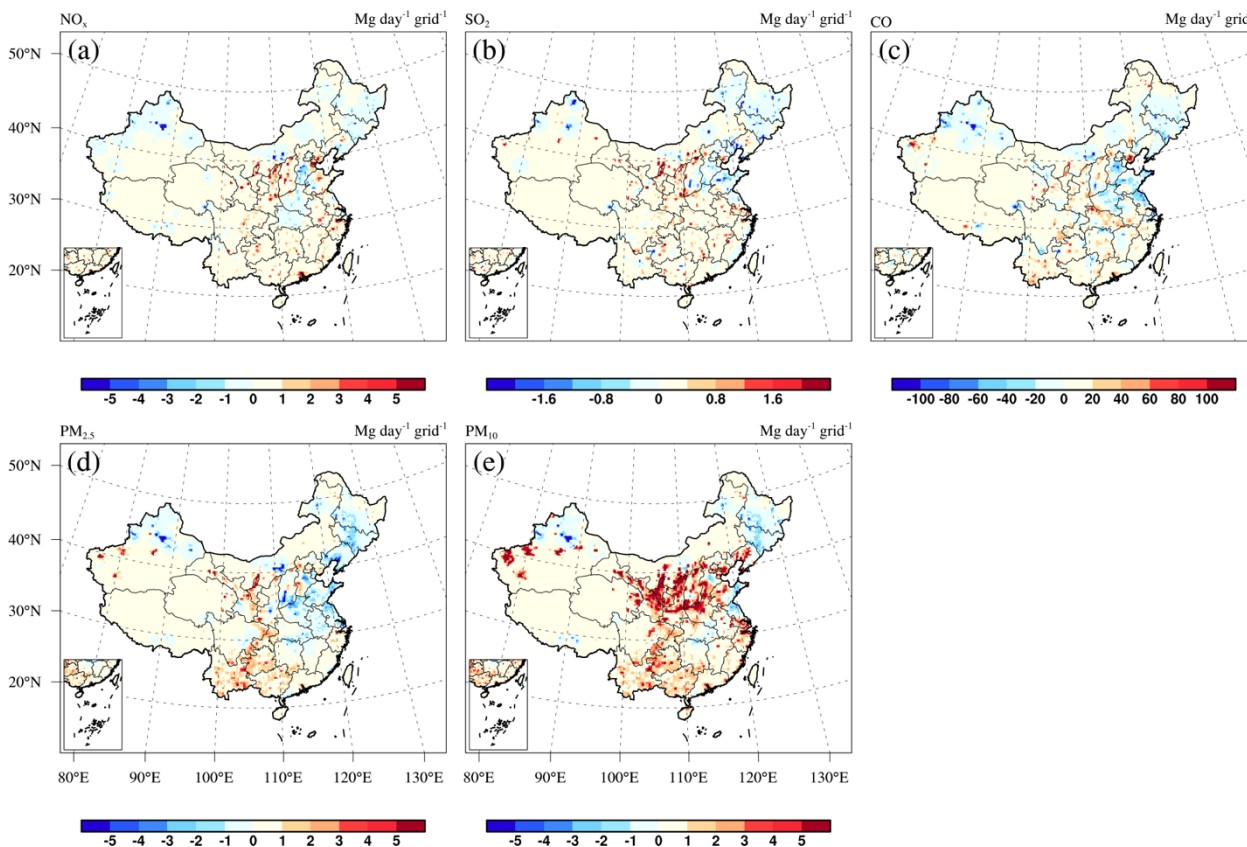


**Figure 7: The inversion estimated emission changes of (a) NO$_x$, (b) SO$_2$, (c) CO, (d) PM$_{2.5}$ and (e) PM$_{10}$ in China from P2 to P3 period.**


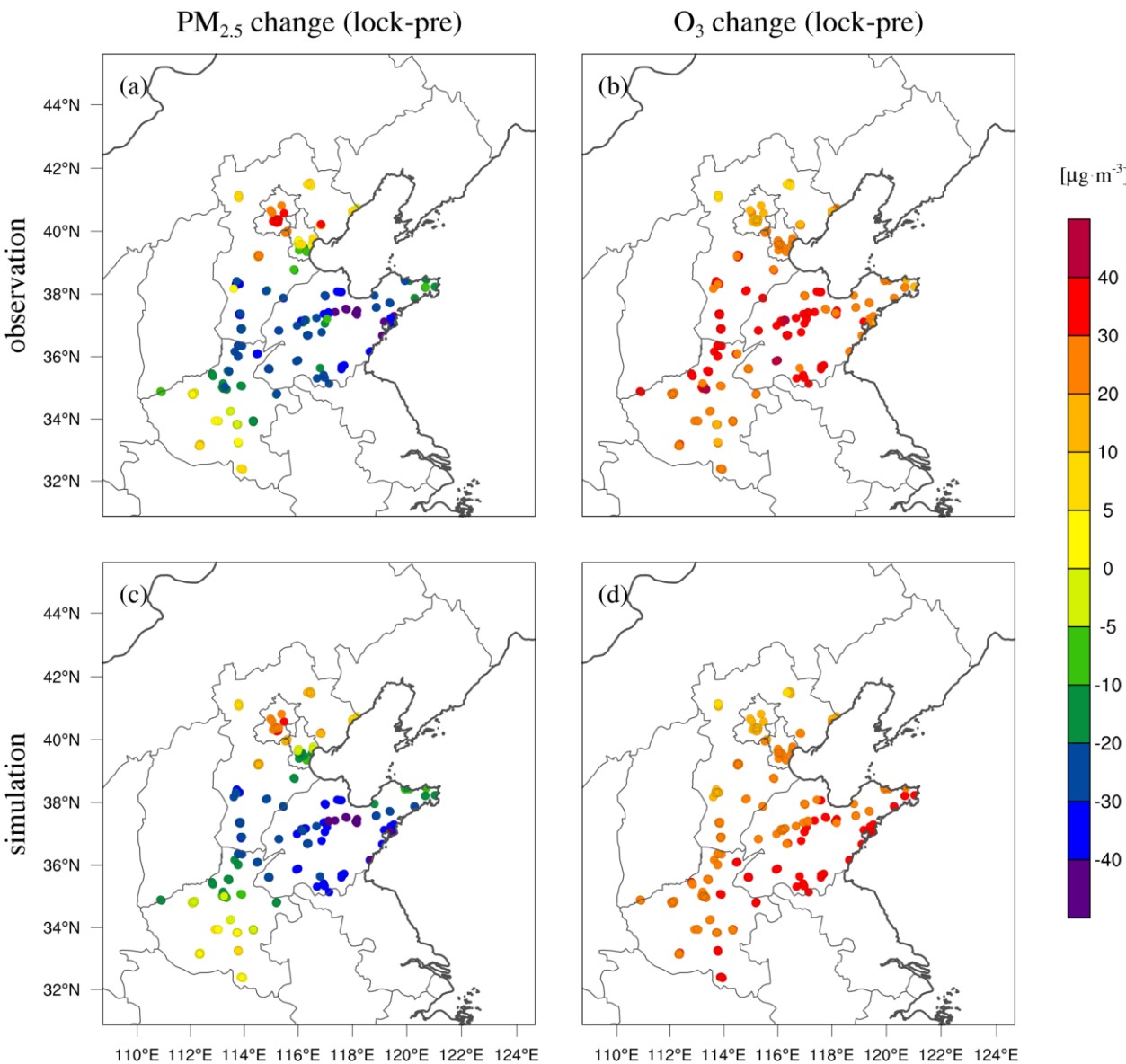


Figure 8: Changes in the observed and simulated concentrations of (a, c) PM₂.₅ and (b, d) O₃ over the NCP region from the pre
lockdown period (P1) to the lockdown period (P2).


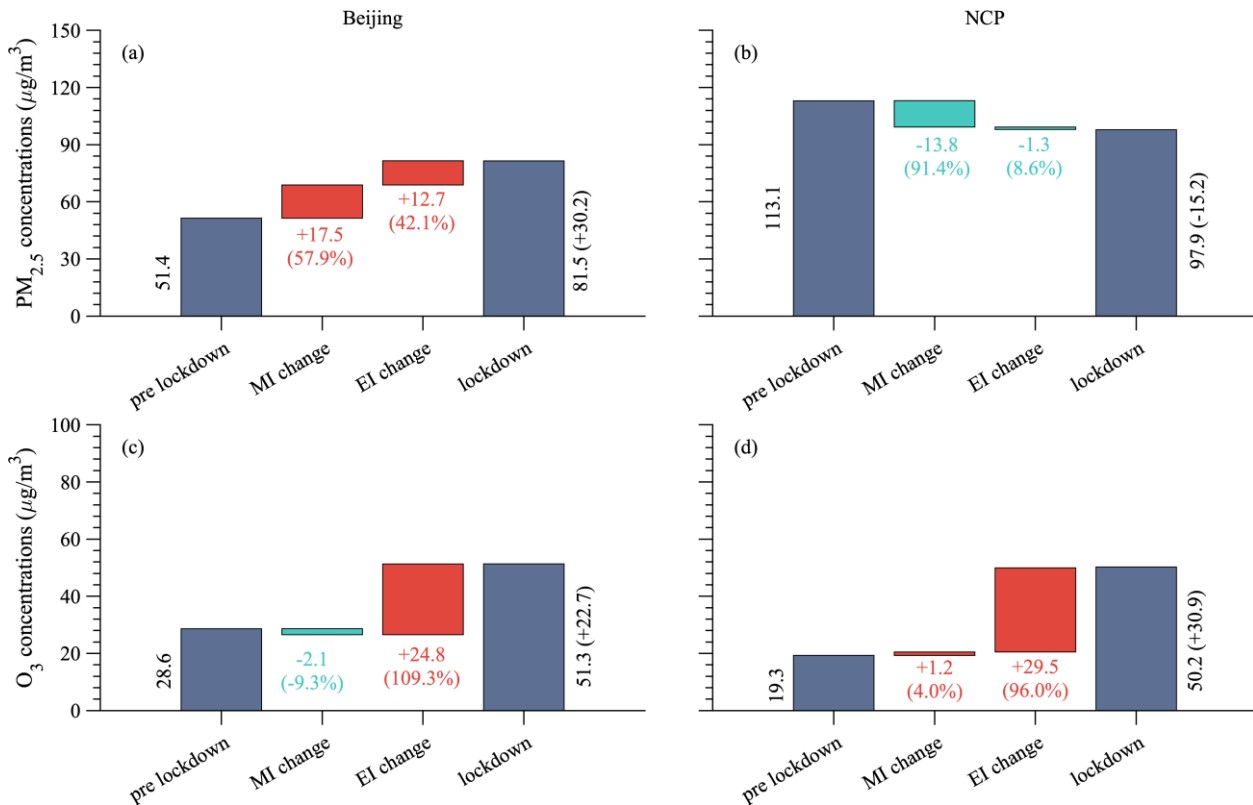

**Figure 9: Contributions of the meteorological variations and emission changes to the changes in (a, b) PM2.5 and (c, d) O3**
**concentrations over Beijing and the NCP region from the pre lockdown period (P1) to the lockdown period (P2).**

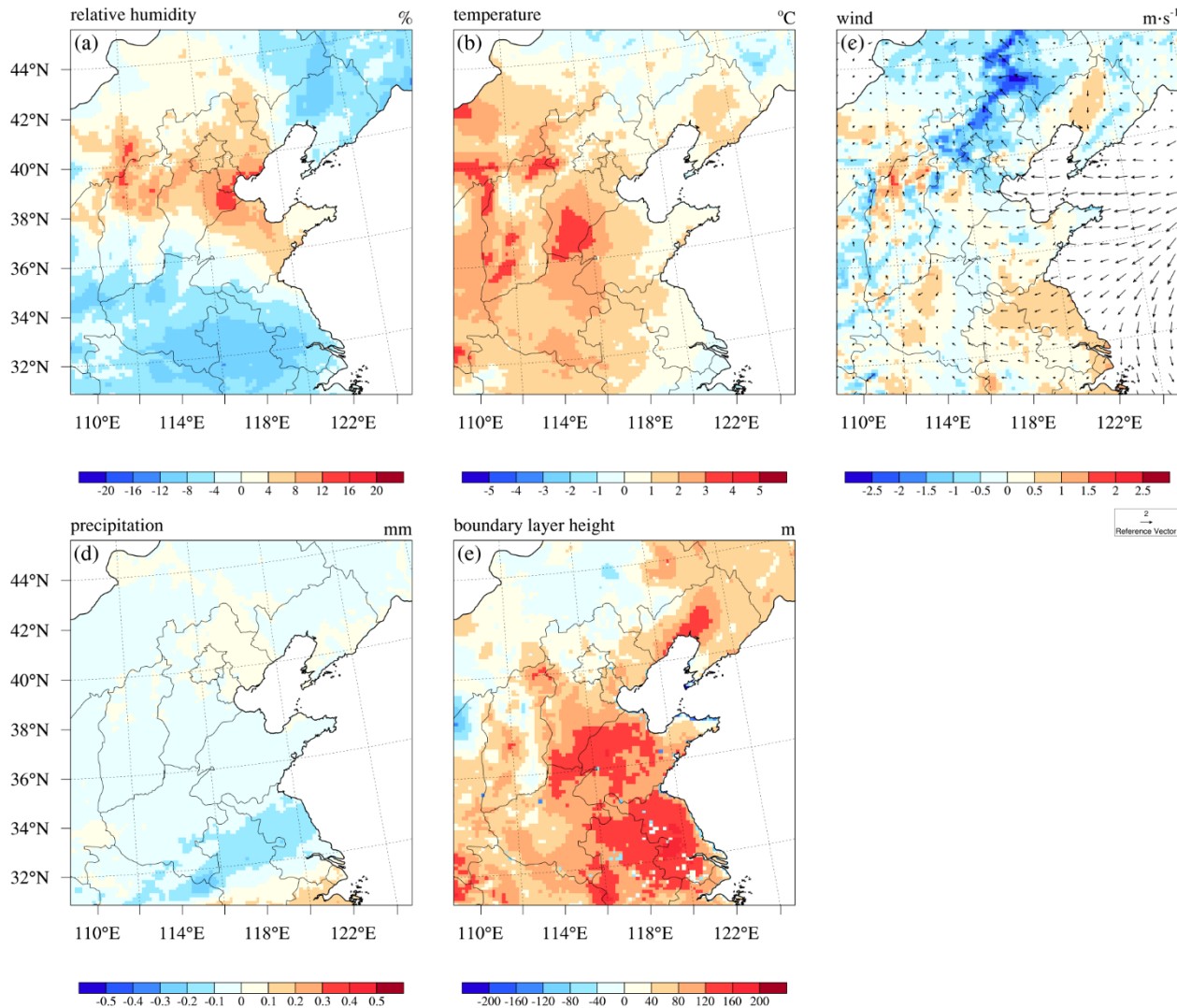

Figure 10: Changes in the (a) relative humidity, (b) temperature, (c) wind speed, (d) precipitation and (e) boundary layer height over the NCP region from P1 to P2 period obtained from WRF simulations.

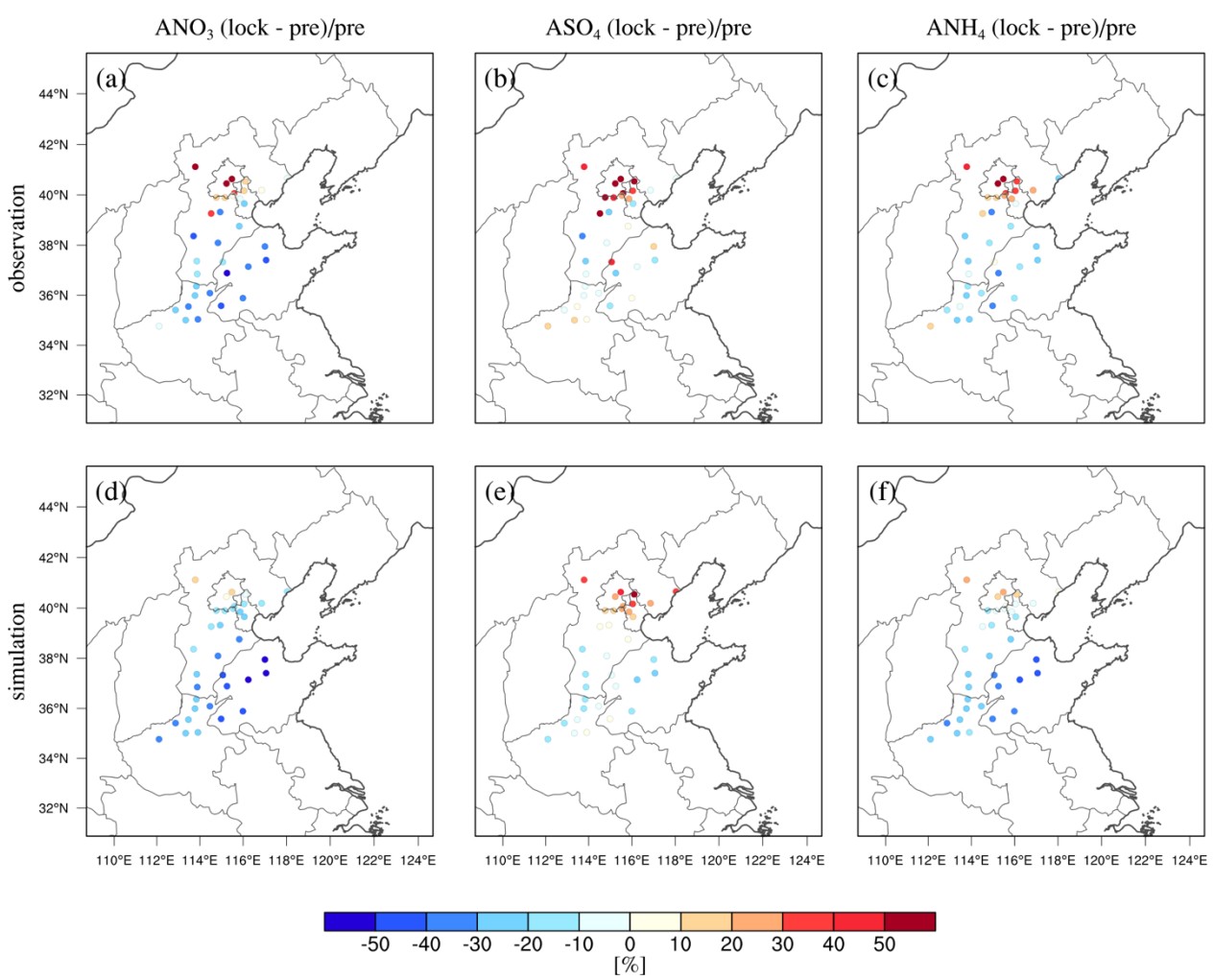


**Figure 11: Relative changes in the simulated and observed concentrations of (a) ANO₃, (b) ASO₄, (c) ANH₄ over NCP region from P1 to P2 period.**



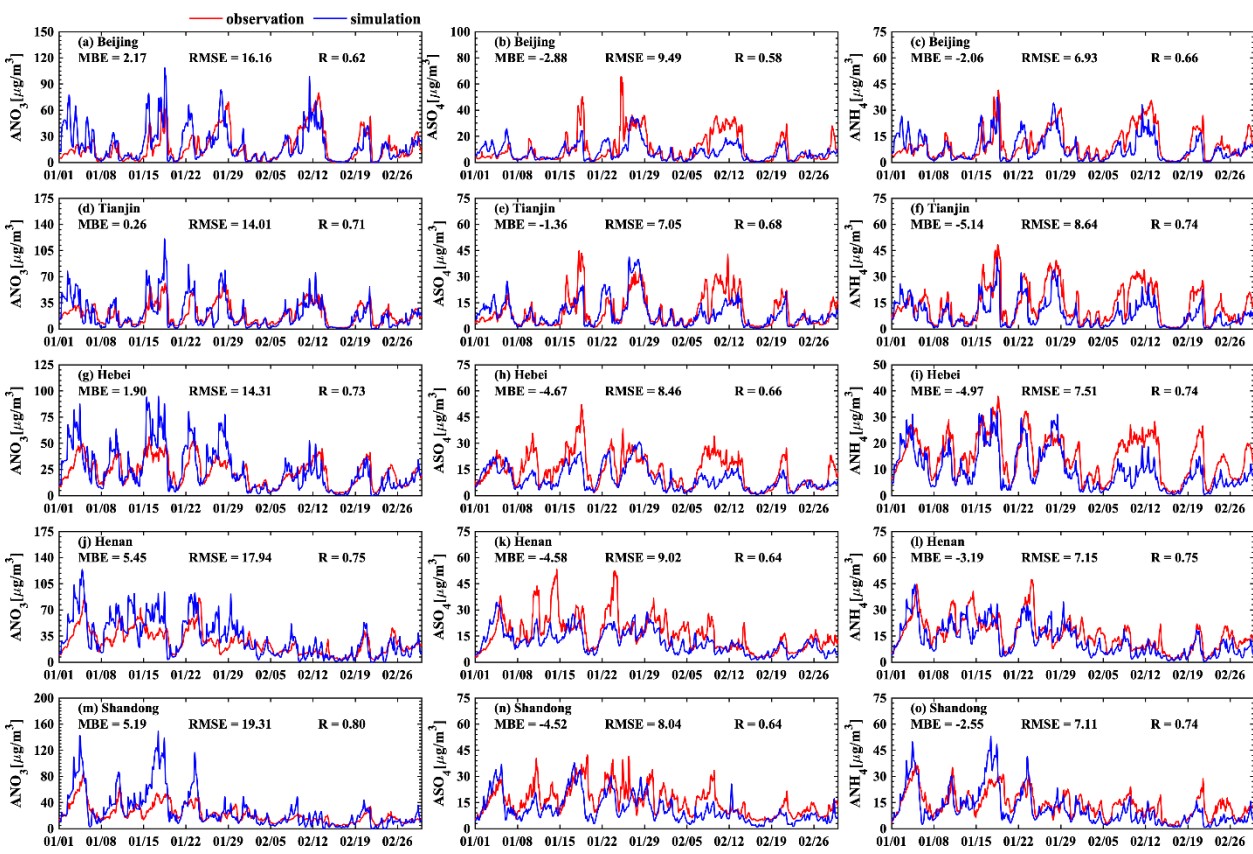


**Figure 12: Time series of observed and simulated concentrations of ANO₃, ASO₄ and ANH₄ in (a-c) Beijing, (b-f) Tianjin, (g-i) Heibei, (j-l) Henan and (m-o) Shangdong province from 1st January to 29th February 2020.**




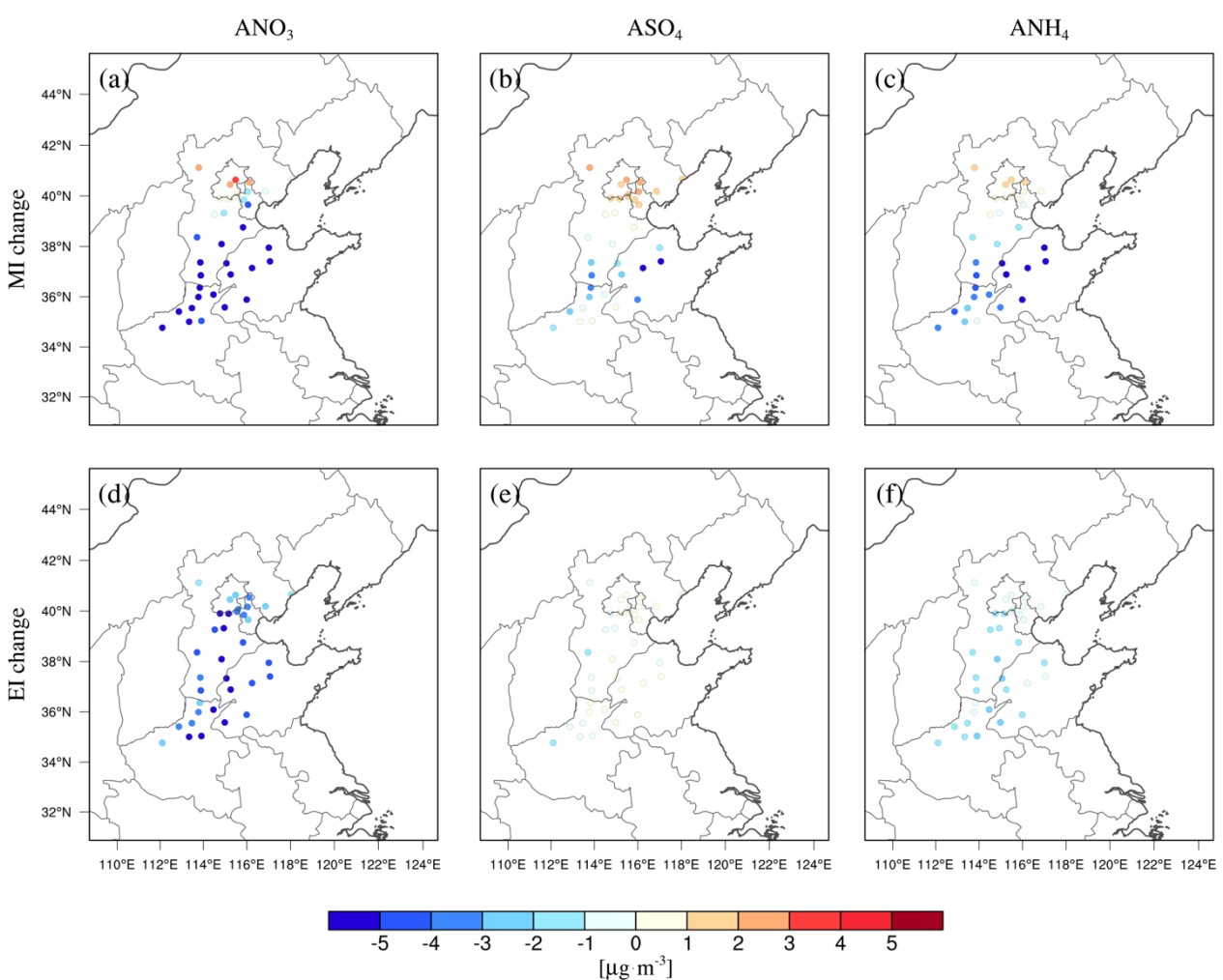


**Figure 13: Meteorology-induced (MI) changes in the concentrations of (a) ANO₃, (b) ASO₄ and (c) ANH₄, as well as Emission-induced (EI) changes in the concentrations of (d) ANO₃, (e) ASO₄ and (f) ANH₄.**



**Data availability**


The hourly surface observations can be obtained from China National Environmental Monitoring Centre (http://www/cnemc.cn/en); The inversion estimated emissions of multi-air pollutants in China during COVID-19 lockdown period and the NAQPMS simulation results are available from the corresponding authors on request.


## Author contributions

X.T., J.Z., and Z.W. conceived and designed the project; H.W., L.K., X.T., and L.W. established the data assimilation system; M.L. Q.W. S.H. W.S. contributed to interpreting the data. L.K. conducted the inversion estimate, drew figures, and wrote the paper with comments provided by J.L., X.P., M.G., P.F., Y.S., H.A. and G.R.C.

## Competing interests

The authors declare no competing financial interest.

## Acknowledgements

We acknowledge the use of surface air quality observation data from CNEMC. This study has been supported by the National Natural Science Foundation of China (grant nos. 41875164, 91644216, 92044303), the CAS Strategic Priority Research Program (grant no. XDA19040201), the CAS Information Technology Program (grant no. XXH13506-302), and the National Key Scientific and Technological Infrastructure project "Earth System Science Numerical Simulator Facility" (EarthLab).

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
