# Peer review of "China during COVID-19 lockdown derived by multi-species surface observation assimilation"

_Atmospheric Chemistry and Physics, 2022_

## Author Comment (AC1)

**Response to Referee #1 (acp-2022-729)**

We Thank Reviewer for his/her constructive comments

Responses to the Specific comments

**General comments:** This work estimated multiple emissions using the EnKF with the state augmentation method during the COVID-19 pandemic. Then, they assessed the unbalanced emission reduction of different species during the restriction. The manuscript is well organized and contains detailed analysis. However, the authors have some issues to be clarified in the manuscript for the final publication in ACP.

**Reply:** The authors appreciate the reviewer for his/her constructive and up-to-point comments. We have carefully considered the comments and revised the manuscript accordingly. Please refer to our responses for more details given below.

**Comment 1:** Line 91 & 285 (Weak motivation): The EnKF coupled with the state augmentation used in this study could be a more advanced method than inversed single estimation (Zhang et al., 2020; 2021, Feng et al., 2020, & Hu et al., 2022) in terms of emission estimation for multiple species. However, it could be less cost-effective. Also, as mentioned in the manuscript (line 285), it shows a similar performance with the single estimation of $SO_2$ (Hu et al., 2022). In other words, the inversed single estimation can be better, considering both performance and computational cost. Therefore, considering these issues, the authors need some more explanation or justify the use of your method to enhance the motivation for this work.

**Reply:** Thanks for this suggestion. The EnKF coupled with the state augmentation is a commonly used method for the estimations of multi-species' emissions (Miyazaki et al., 2017; Ma et al., 2019; Peng et al., 2018). In this method, the emissions of different species are simultaneously estimated by including them as a part of the state vector together with the chemical concentrations using the ensemble model simulations and observations. The mass balanced method used in Zhang et al. (2020) and Zhang et al. (2021) is to adjust the model emissions based on the modeled local ratio between concentrations and emissions. Although it has lower computational cost than the EnKF method since it does not need to run an ensemble of model simulation, it has difficulties in accounting for the nonlinear relationship between the concentrations and emissions, thus is more commonly used in the inversions of short-lived species (e.g., $NO_x$) under relatively coarse (>1°) resolution (Streets et al., 2013). The EnKF method instead can consider the nonlinear relationship between the emissions and concentrations (as we illustrated in our response to comment 2), and can be used for longer-lived species (e.g., $SO_2$, $PM_{2.5}$ and CO) at finer model

resolutions (Streets et al., 2013). The four-dimensional variational assimilation (4DVar) method used in Hu et al. (2022) is an advanced inversion method similar to the EnKF method, and they have different strengths and weakness (Skachko et al., 2014). The 4DVar uses an adjoint model of a chemical transport model (CTM) to estimate the emissions, thus it requires the development and maintenance of an adjoint model, which is technically difficult and cumbersome for the complex CTM. The EnKF method does not require an adjoint model and is easily implemented. Besides, although 4DVar does not need to run an ensemble of simulation, it needs to solve a complex optimization problems for a large chemical model system, therefore, the 4DVar have comparable computational costs with the EnKF method (Skachko et al., 2014). Similar to our method, Feng et al. (2020) used the EnKF method to estimate the emission changes of $NO_x$ during COVID-19 pandemic, but they did not constrain the emissions of other species. Due to the chemical reactions in the atmosphere, the concentrations of different species are interrelated with each other. For example, the ambient $PM_{2.5}$ is not only primarily emitted, a large portion of them is also formed secondarily through reactions with several gaseous precursors, such as $NO_2$ and $SO_2$. This means that inversed single estimation may cause biases in the inversed $PM_{2.5}$ emission if the errors in emissions of $NO_2$ and $SO_2$ were not corrected synchronously. Assimilations of CO measurement also influences the inversion of $NO_2$ emission due to its influence on the OH concentration in the atmosphere (Miyazaki and Eskes, 2013). Therefore, doing the multi-species inversion estimation can provide more constraints on the atmospheric chemical system than the inversed single estimation and thus can produce more reasonable inversion results (Miyazaki and Eskes, 2013; Peng et al., 2018). The inversion estimation for $SO_2$ may be less affected by assimilations of other species due to its less dependence on other species. As a result, our inversion results showed a similar performance with the single estimation of $SO_2$ in Hu et al. (2022). Besides, the emissions of different species were perturbed simultaneously in our method, thus the ensemble simulation only needs to be performed once. The computational cost only increased slightly in the analysis step. Therefore, the multi-species' inversion method used in our study has its strength over the inversed single estimation method used in previous studies. Following the suggestions of reviewers, we give more explanation for the use of our method in the revised manuscript (please see lines 161–178)

**Comment 2:** Line 126: The authors need to clarify how to consider the non-linearity between NOx emission (for estimation) and $NO_2$ concentration (of observation). Additionally, in-situ $NO_2$ observation may be based on the commercial chemiluminescent instrument. While the instrument system converts $NO_2$ to NO through a molybdenum converter, other species, such as peroxyacetyl nitrates (PANs) and

HNO3, are also simultaneously converted to NO. The other species account for a large portion of the converted NO molecules. For example, Dunlea et al. (2007) showed the interference in the chemiluminescence detection accounting for up to 50% of ambient $NO_2$. In other words, the observed $NO_2$ is almost equivalent to ambient $NO_y$. Also, PAN is thermally sensitive (and it is also related to the temperature shown in Figure 10). The rapid decrease in $NO_x$ emission is strongly linked to this issue. Therefore, the authors also need to explain how to treat these issues in your calculation.

Reply: Thanks for this important comment. In EnKF method, the relationship between the concentrations and emissions of related species was determined by the background error correlations, which is estimated from the ensemble model simulations. Since the chemistry transport model used in our study is a nonlinear model which considers the physical and chemical processes in the atmosphere, the ensemble simulation is able to represent the nonlinear evolution of the error correlations. This allows the nonlinear relationships between $NO_2$ concentrations and $NO_x$ emissions to being considered through the use of background error covariance produced by ensemble simulations (Evensen, 2009; Miyazaki et al., 2012). Following the suggestions of reviewer, we have clarified this in the revised manuscript (please see lines 165–167).

As the reviewer mentioned, the $NO_2$ measurement made by CNEMC is based on the chemiluminescent analyser with a molybdenum converter. We agree with the reviewer that the interference of $HNO_3$, PAN and alkyl nitrates (AN) can lead to an overestimation of $NO_2$ (Dunlea et al., 2007; Lamsal et al., 2008) and may lead to spurious decreases of $NO_x$ emissions. Due to the lack of synchronous observations of $HNO_3$, PAN and AN, it is hard to directly correct such overestimations in the $NO_2$ measurements. Previous studies (Cooper et al., 2020; He et al., 2022) usually use chemical transport model to simulate $NO_x$, $HNO_3$, PAN and AN to produce correction factors (CFs) for the $NO_2$ measurements using the following relationship proposed by Lamsal et al. (2008):

$$CF = \frac{[NO_2]}{[NO_2] + 0.95[PAN] + 0.35[HNO_3] + \sum[AN]} \tag{R1}$$

but the calculation of CF could be affected by the simulation errors in the model caused by uncertainties in emission inventory or other error sources, which may contaminate the observations. Therefore, similar to Feng et al. (2020), we did not correct the $NO_2$ measurement in our inversion of $NO_x$ emissions since there were large uncertainties in the $NO_x$ emissions during the COVID-19 pandemic that possibly led to erroneous CF. Instead, the EnKF is capable of considering the errors in the observations during the assimilation through the use of observation error covariance matrix. The interference in the chemiluminescence detection to the $NO_2$ measurement was thus treated as a kind of observation error in

our study. According to our estimates, the total observation error of $NO_2$ measurements was about 6–10 µg/m$^3$. To investigate whether our settings of observation error can fully cover the errors induced by the chemiluminescence monitor interference, we calculated the CF according to Eq. (R1) based on the simulated $NO_2$, $HNO_3$, PAN and AN using the inversed emission inventory to alleviate the effects of emission uncertainty on the CF calculations. Figure R1 shows the calculated CFs for $NO_2$ measurements over different regions of China during COVID-19 pandemic, which generally ranged from 0.75 to 0.99. The CF values over NCP, NE, NW and Central were generally stable throughout the COVID-19 pandemic, all larger than 0.9, suggesting that chemiluminescence monitor interference only has slight effects on the $NO_2$ measurement. Over the SE and SW regions, there was a drop of CF values during the lockdown period, followed by an increase after the lockdown. This indicates that the decline of $NO_2$ concentrations during lockdown period may be larger in these two regions.

[Figure]

**Figure R1: Time series of calculated CFs for NO₂ measurements over (a) NCP, (b) NE, (c) SE, (d) SW, (e) NW and (f) Central region during COVID-19 pandemic. The averaged CF values during different stages of COVID-19 pandemic are also labeled.**

As shown in Figure R2, the overestimations were generally lower than 3 $µg/m^3$ over different regions of China throughout the COVID-19 pandemic, which is smaller than the observation errors we assigned in the assimilation, suggesting that the observation error caused by the chemiluminescence monitor interference were contained in the assimilation. In order to quantify the influences of chemiluminescence monitor interference on the inversed $NO_x$ emission, an additional inversion

experiment was conducted based on the corrected $NO_2$ measurement using the calculated CF. The results suggest the chemiluminescence monitor interference in the $NO_2$ observations had weak impacts on the inversed $NO_x$ emissions as seen in Fig. R3 and Fig. R4, which display the comparisons of the inversed $NO_x$ emission with and without correction in respect of the magnitude and change ratio during different stage of COVID-19 pandemic. The differences in the magnitude of inversed $NO_x$ emissions caused by correction were about 2–7% over the NCP, NE, NW and Central, and were about 10–13% over the SE and SW. Differences in the emission reductions of $NO_x$ were also small, which was about 0.3 to 4.1 percentage points. Consistent with the reviewer, the result suggests that the chemiluminescence monitor interference is a potential factor that influence the inversion of NOx emission, which is a limitation of current work. However, it might not significantly influence our inversion results and the main conclusions. Considering this, we added a discussion about the effects of chemiluminescence monitor interference to inform the potential reader in the revised manuscript (please see lines 143–158) and supplement (please see lines 15–39 and Figure S16–S19).

[Figure]

**Figure R2: the difference of NO₂ measurement before and after the corrections of chemiluminescence monitor interference over (a) NCP, (b) NE, (c) SE, (d) SW, (e) NW and (f) Central during the COVID-19 period.**

[Figure]

**Figure R3 Comparisons of inversed NOₓ emissions with (blue) and without (red) correction of NO₂ measurement over different regions of China during different period of COVID-19 pandemic.**

[Figure]

**Figure R4 Comparisons of the calculated emission change of NOₓ emissions based on the inversion results with (blue) and without (red) correction of NO₂ measurement over different regions of China.**

**Comment 3**: Line 90: Levelt et al. (2022) is not related to the emission inversion technique. The authors had better discard this paper in the manuscript.

Reply: We feel sorry for this inappropriate citation. We have removed this paper in the revised manuscript.

**Comment 4:** Line 105: Anthropogenic and other emission inventories used in the simulations during the COVID-19 pandemic are based on 2010 or relatively long ago. These emission rates could be significantly higher than those during the COVID-19 pandemic. The higher emission rates are significantly related to the concentration of atmospheric species like $O_3$. Therefore, the authors need to justify the uses of such emission inventories in the simulations

**Reply:** Thanks for this important comment. We agree with the reviewer that a prior emission inventory used in our study is significantly higher than those during the COVID-19 pandemic due to the restrict emission control policy over the past decades. However, due to the lack of updated activity data and emission factors, the latest bottom-up anthropogenic emission inventory in China that were available is for the base year 2018, which could be also higher than those during COVID-19 pandemic. Considering this, we have developed the iteration emission inversion technique to address this issue. According to the inversion results, the iteration emission inversion significantly reduced the large biases in the a priori emission inventory and is able to reproduce the emission levels during the COVID-19 pandemic. Therefore, we did not update the a priori emission inventory to more recent emission inventory during the assimilation.

To test the influences of the choice of a priori emission inventory on the inversion estimation, a new inversion run was conducted based on the a priori emission inventory for base year 2018. This new emission inventory is comprised of the anthropogenic emissions obtained from HTAPv3 (Crippa et al., 2023), the biogenic, soil and oceanic emissions obtained from the CAMS global emission inventory (https://ads.atmosphere.copernicus.eu/cdsapp#!/dataset/cams-global-emission-inventories?tab=overview, last access: March 15, 2023) and biomass burning emissions obtained from the Global Fire Assimilation System (GFAS)(Kaiser et al., 2012). The detailed steps of the new inversion estimation were same as those elucidated in Sect.2. Figure R5–7 show the comparisons of inversion results based on the a priori emissions for base 2010 with those based on the a priori emissions for 2018. The results suggest that the inversion results based on the 2010 and 2018 inventory were broadly close to each other, while the inversion results based on 2018 inventory were relatively higher than those based on 2010 inventory, reflecting the uncertainty in our inversion results caused by the choice of the a priori emission inventory. Figure R8 and R9 show the temporal variations of the multi-species' emissions during COVID-19 pandemic at the national and regional scales derived from the inversion results based on 2018 inventory, which consistently showed that the $NO_x$ emissions decreased much larger than other species. In all, the sensitivity experiment demonstrates that the choice of a priori emission inventory may not obviously

influence the main conclusion of our study, but can lead to uncertainty in the magnitude of the inversion results which is a limitation of current work. Therefore, a discussion about the influence of a priori emission inventory has been added to the revised manuscript (please see lines 492–505) and supplement (please see Figure S20–S24) to inform the potential readers.

[Figure]

**Figure R5: Comparisons of the inversion estimated total emissions of (a) NO$_x$, (b) SO$_2$, (c) CO, (d) PM$_{2.5}$ and (e) PM$_{10}$ before lockdown using the a priori emission inventory for base year 2010 (read) with those using the a priori emission inventory for base 2018 (blue).**

[Figure]

**Figure R6: Same as Fig.R5 but for the p2 period (lockdown period).**

[Figure]

**Figure R7: Same as Fig. R5 but for the P3 period (after back-to-work day).**

23rd January    3rd February 10th February

P1 (before lockdown)    P2 (lockdown)    P3 (after back-to-work day)

**Figure R8: Time series of normalized emission anomalies estimated by inversion results for different species in China from 1st January to 29th February 2020 using the a priori emission inventory for 2018.**

[Figure]

**Figure R9: Time series of normalized emission anomalies estimated by inversion results for different species over (a) NCP region, (b) NE region, (c) SE region, (d) SW region, (e) NW region and (f) Central region from 1st January to 29th February 2020 using the a priori emissions for 2018.**

**Comment 5**: Line 138: The explanation of EI and MI (from both MET and EMIS change scenarios) is rather complicated. The authors need to clarify it. Tabulating or illustrating the scenarios or cases makes it easier for the readers to understand them.

**Reply:** Thanks for this suggestion. We have added two tables in the revised manuscript to illustrate the different scenarios and the meaning of different items in the calculation of EI and MI (please see Table 2 and Table 3 in the revised manuscript):

**Table R1 configuration of simulation scenarios**

| Scenarios | Meteorology input | Emission input | Purpose |
|---|---|---|---|
| BASE scenario | varied meteorological condition from pre lockdown to lockdown period | varied emission from pre-lockdown to lockdown period | To estimate the total changes of air pollutant concentrations induced by emission and meteorological change |
| MET change scenario | varied meteorological condition from pre-lockdown to lockdown period | constant emissions during pre-lockdown and lockdown period | To estimate the impacts of meteorological changes on the air pollutants |
| EMIS change scenario | constant meteorological during pre-lockdown and lockdown period | varied emission from pre-lockdown to lockdown period | To estimate the impacts of emission changes on the air pollutants |

**Table R2 descriptions of different items used in the calculation of meteorological-induced and emission-induced changes of air pollutant concentrations**

| notation | Description |
| --- | --- |
| $MI$ | meteorological-induced changes of air pollutant concentrations |
| $EI$ | emission-induced changes of air pollutant concentrations |
| $MI_{MET\ change\ scenario}$ | meteorological-induced changes of air pollutant concentrations calculated by the MET change scenario |
| $EI_{MET\ change\ scenario}$ | emission-induced changes of air pollutant concentrations calculated by total changes minus $MI_{MET\ change\ scenario}$ |
| $EI_{EMIS\ change\ scenario}$ | emission-induced changes of air pollutant concentrations calculated by the EMIS change scenario |
| $MI_{EMIS\ change\ scenario}$ | meteorological-induced changes of air pollutant concentrations calculated by total changes minus $EI_{EMIS\ change\ scenario}$ |
| $conc_{p1,BASE\ scenario}$ | averaged concentrations of air pollutants during P1 period under the BASE scenario |
| $conc_{p2,BASE\ scenario}$ | averaged concentrations of air pollutants during P2 period under the BASE scenario |
| $conc_{p1,MET\ change\ scenario}$ | averaged concentrations of air pollutants during P1 period (pre-lockdown) under the MET change scenario |
| $conc_{p2,MET\ change\ scenario}$ | averaged concentrations of air pollutants during P2 period (lockdown) under the MET change scenario |
| $conc_{p1,EMIS\ change\ scenario}$ | averaged concentrations of air pollutants during P1 period under the EMIS change scenario |
| $conc_{p2,EMIS\ change\ scenario}$ | averaged concentrations of air pollutants during P2 period under the EMIS change scenario |
| $contri_{met}$ | relative contributions (%) of the meteorological variations to the changes in air pollutant concentrations |
| $contri_{emis}$ | relative contributions (%) of the emission changes to the changes in air pollutant concentrations |

**Comment 6**: Line 218: The authors need to define P1, P2, and P3 on Line 218, not on Line 247. Also, the authors need to add some lines for P1, P2, and P3 on the x-axis of Figure 4.

**Reply:** Thank you so much for your careful check. We have changed the position where the P1, P2 and P3 were defined in the revised manuscript (please see lines 104–106) and some lines for P1, P2 and P3 were also added on the x-axis of Figure 4 and Figure 5.

**Comment 7:** Line 268: The performance in the $O_3$ simulation is relatively poor. As mentioned in the manuscript, the lower performance is related to VOC emission. Since the VOC emission used in the simulations is made for 2010, there are some time gaps. Therefore, the authors need to compare the VOC emission rates with the recent or 2019 database and then make an additional explanation of (expected) ozone concentrations.

**Reply:** Thanks for this comment. Following the suggestions of reviewer, we have compared the NMVOC emissions for base year 2010 with the NOVOC emissions for base year 2018. To prevent the inconsistency between different inventory, the anthropogenic part of NMVOC emissions were obtained from the HTAP_v3 inventory(Crippa et al., 2023), which is an updated version of the anthropogenic emission inventory (HTAP_v2.2) we used in our study. The biogenic part was obtained from the CAMS biogenic emissions calculated using the Model of Emissions of Gases and Aerosols from Nature (MEGAN) driven by ERA-Interim meteorological fields (Granier, C. et al., 2019). The NMVOC emissions from wildfires and biomass burning was obtained from the Global Fire Assimilation System (GFAS)(Kaiser et al., 2012). Figure R5 shows the comparisons of NMVOC emissions for base year 2010 with those for base year 2018 over different regions of China. It shows that the NMVOC emissions for base year 2010 were generally lower than those for 2018 except over the SW regions. Considering the increasing trend of NMVOC emissions in China (Li et al., 2019), the underestimates of NMVOC emissions for base year 2020 due to the use of old emission inventory may be larger. This is in line with the negative biases in the simulated $O_3$ concentrations over these regions. Following the suggestion of reviewer, we have clarified it in the revised manuscript (please see lines 294–297).

[Figure]

**Figure R5 comparisons of the NMVOC emissions for base year 2010 with those for 2018 over different regions of China.**

**Comment 8:** Line 289 & 303 – 304: The authors need to explain what causes increases in $PM_{10}$ during the P3 in Figure 4, Figure 5e, and 5f. The authors explained these are related to the sandstorm. However, the concentration of $PM_{10}$ during the P3 period was rather low in the NW and Central regions.

**Reply:** Thanks for this suggestion. The $PM_{2.5}/PM_{10}$ ratio was used to investigate the causes of the increases in $PM_{10}$ emission in the revised manuscript, which is an indicator of the potential sources of particular matter. A lower $PM_{2.5}/PM_{10}$ ratio usually indicates significant contributions from natural sources such as dust (Wang et al., 2015; Fan et al., 2021). As we can see from Fig.R6, the $PM_{2.5}/PM_{10}$ ratio was stable during the P1 and P2 period, but it decreased substantially during the P3 period, from 0.81 to 0.48 over the NW region and from 0.77 to 0.53 over the Central region, which suggests larger contributions of dust emissions to the PM10 concentrations during the P3 period. Moreover, the NW and Central region are typical source areas of dust in China, therefore the increasing of $PM_{10}$ emissions over NW and Central regions may be mainly related to the enhanced dust emissions. following the suggestion of reviewer, we added more explanations to the increased $PM_{10}$ emissions over the NW and Central region in the revised manuscript (please see lines 342–348).

[Figure]

**Figure R6 timeseries of PM$_{2.5}$/PM$_{10}$ ratio during COVID-19 pandemic over (a) NW and (b) Central region**

**Comment 9:** Line 295: The explanation is not sufficient for the PM$_{2.5}$ emission increase in the NCP region. The observation done by Dai et al. (2020) was carried out at a single site in Tianjin.

**Reply:** Thanks for this important comment. We conducted a more detailed analysis on the possible causes of the PM$_{2.5}$ emission increases in the NCP region through literature review and analysis of the PM$_{2.5}$ compositions. Previous researches suggested that the increases of PM$_{2.5}$ emissions over the NCP region may be due to the increased emissions from industry and fireworks (Dai et al., 2020; Li et al., 2021; Ma et al., 2022; Zuo et al., 2022). Based on the measurement of stable Cu and Si isotopic signature and distinctive metal ratios in Beijing and Hebei, Zuo et al. (2022) analyzed the variations in the PM$_{2.5}$ sources in Beijing and Hebei during the COVID-19 pandemic, which provides evidences that the primary PM$_{2.5}$ emissions did not decrease in Beijing and Hebei, and that the PM-associated industrial emissions may instead increase during the lockdown period. The increased industrial heat sources detected by Li et al. (2022) based on VIIRS active fire data also supported the increased industrial emissions over the NCP region during lockdown period. Figure R7 shows the variations of the concentrations of potassium (K$^+$) and magnesium (Mg$^{2+}$) ion, two important fingerprints of the firework emissions, over the NCP region during COVID-19 pandemic. Measurement of K$^+$ and Mg$^{2+}$ over the NCP region were obtained from China National Environmental Monitoring Center (CNEMC) with site distribution shown in Fig. 11. Substantial increases of K$^+$ and Mg$^{2+}$ concentrations could be observed during the Spring Festival over

the NCP region, which indicates larger contributions of firework emissions to the $PM_{2.5}$ concentrations during the lockdown period. this is consistent with the field measurements in Beijing and Tianjin conducted by Ma et al. (2022) and Dai et al. (2020). These results suggested that the increased industrial $PM_{2.5}$ emissions, together with firework emissions may contribute to the increased $PM_{2.5}$ emissions over NCP region, which compensated the emission reductions from the traffic emissions. Based on this analysis, we have added more detailed discussions about the possible reason for the increases of $PM_{2.5}$ emissions over the NCP region in the revised manuscript (please see lines 322–333) and supplement (please see Figure S15).

[Figure]

**Figure R7: Timeseries of averaged concentrations of potassium and magnesium ion during COVID-19 pandemic over the NCP region.**

**Comment 10:** Line 340: The authors also need to mention that the $O_3$ is under-simulated in all regions (refer to Figure S6).

**Reply:** Thanks for this comment. The underestimations of the $O_3$ have been pointed out in the revised manuscript (please see lines 389).

**Minor comments:**

**Comment 11:** Line 32: measurement -- > measure or measures

**Reply:** Done.

**Comment 12:** Line 33: remove 'both'.

**Reply:** Done.

**Comment 13:** Line 261. It is probably Table S1, not Table S4

**Reply:** Done.

**Comment 13:** Lines 277 and 280: The authors need to confirm the numbers in these lines (and Table 2).

**Reply:** Thanks a lot for your careful check. We have corrected the wrong number in lines 277 and 280. Please see lines 305 and 308 in the revised manuscript.

---

## Author Comment (AC2)

**Response to Referee #2 (acp-2022-729)**

We Thank Reviewer for his/her constructive comments

Responses to the Specific comments

**General comments:** This manuscript evaluates the changes in air quality during the COVID-19 lockdown by dividing them into meteorological and emissions parts. The authors specifically assessed the impact of emission reduction during the lockdown using a multi-air pollutant inversion system and observational data, which is a unique approach compared to previous studies. The findings from this study will be valuable in evaluating the effectiveness of emission reduction policies by policymakers in polluted regions, including China. However, to be published in ACP, the authors must address the following issues:

**Reply:** The authors appreciate the reviewer for his/her constructive suggestions. In the revised manuscript we have considered each comment for improvement, revision, and correction. Please refer to our responses for more details given below.

**Comment 1:** Line 39: During the COVID-19 period, a haze event also occurred. However, the use of the term "COVID-19 haze" may convey the notion that the pandemic was the cause of the haze phenomenon. Thus, the authors should choose terms more carefully.

**Reply:** Thanks for pointing out this issue. The "COVID-19 haze" has been replaced by "unexpected $PM_{2.5}$ pollution during the COVID-19 lockdown" in the revised manuscript.

**Comment 2:** Lines 128-129: Despite removing unrealistic observations through the Wu et al. (2018) method, some extreme values persist in the time-series plots (e.g., Figures 3(b), S1(b), and S2(b)). Hence, the authors should thoroughly verify that the raw data has been properly filtered.

Reply: Thanks for this important comment. We have checked the quality of observation data we used in the assimilation. According to Fig. 3(b), S1(b) and S2(b), the possible extreme values for $PM_{2.5}$, $PM_{10}$ and $SO_2$ mainly occurred over the NE region around 25th January. Figure R1(a) shows the spatial distributions of daily averaged $PM_{2.5}$ concentrations over the NE region. Obvious high $PM_{2.5}$ concentrations could be found at multiple monitoring sites over Liaoning province, where the $PM_{2.5}$ concentrations increased sharply (over 400 $\mu g/m^3$) at the night of 24th January and early morning of 25th January (Fig. R1(b)). Same phenomena also occurred in the city Baotou at Inner Mongolia (Fig.R1(c)). Considering that the 24th January was the 2020 Chinese New Year Eve and there are traditions of setting fireworks at the night of that day, the peak $PM_{2.5}$ concentrations at the night of 24th January may be related

to the firework emissions. Meanwhile, there were multiple sites that showed same signals of high PM$_{2.5}$ concentrations. Thus, we think the high PM$_{2.5}$ concentrations during 24$^{th}$ January is reasonable, and did not treat them as outliers. Similar conclusions can be drawn for PM$_{10}$ and SO$_2$, as seen in Fig. R2 and R3.

[Figure]

**Figure R1: (a) spatial distribution of daily averaged PM$_{2.5}$ concentrations over the NE region at 25$^{th}$ January 2020, and the time series of averaged PM2.5 concentrations at (b) Liaoning province and (c) the city of Baotou at Inner Mongolia from 24$^{th}$ to 26$^{th}$ January 2020.**

[Figure]

**Figure R2: Same as Fig. R1 but for PM$_{10}$ concentrations**

[Figure]

**Figure R3: Same as Fig. R1 but for SO₂ concentrations**

**Comment 3**: Lines 208-242: (Section 2.4) The authors assess both the MI and EI approaches for decreasing nonlinear effects. If the extent of nonlinearity (or sensitivity) demonstrated by the two methods is documented in the paper, it can provide a helpful reference for future research.

**Reply:** Thanks for the suggestion. We analyzed the differences between the MI and EI method in the revised manuscript (please see lines 393–398) and supplement (please see Figure S13 and S14). According to Figure R4 and R5, the calculated MI and EI changes of PM$_{2.5}$ and O$_3$ concentrations were consistent with each other over the Beijing and the NCP region, and indicates similar conclusions. The differences of calculated MI and EI were within 2 $\mu g/m^3$ for PM$_{2.5}$ concentrations, which were small in this application. In terms of O$_3$ concentrations, the differences were larger, which were around 5 $\mu g/m^3$ over the Beijing and NCP region. In addition, the sign of calculated MI using EMIS change scenario and MET change scenario were opposite although both suggested weak contributions of meteorological variation to the changes of O$_3$ concentrations, suggesting that the calculated MI and EI changes of O$_3$ concentrations could be more sensitive to the used scenarios. This may be due to the stronger nonlinearity of O$_3$ concentrations to the meteorology and emissions.

[Figure]

**Figure R4: The calculated MI and EI changes of PM2.5 concentrations over the (a, c) Beijing and (b, d) the NCP region using the EMIS change scenario (upper panel) and MET change scenario (lower panel).**

[Figure]

**Figure R5: Same as Fig. R4 but for O3 concentrations.**

**Comment 4:** Lines 302-304: The authors posit that the rise in $PM_{10}$ emissions in the NW and central regions during P3 is due to sandstorms but do not provide clear evidence. Furthermore, the simulation using a priori in the central region does not show a significant deviation from observation. Thus, the authors must provide further evidence for the sandstorm hypothesis.

**Reply:** Thanks for raising this issue. As we illustrated in Sect. 2.3 (lines 215–217 and table 1), the $PM_{10}$ emissions were calculated by the sum of the emissions of $PM_{2.5}$ and PMC (coarse mode unspeciated aerosol) which were respectively constrained by the concentrations of $PM_{2.5}$ and $PM_{10-2.5}$. As we can see from Fig. R6, there were significantly larger deviations in the simulated $PM_{10} - PM_{2.5}$ concentrations from the observations over the Central region during the P3 period, which led to the rise in $PM_{10}$ emissions over there. However, as the reviewer mentioned, the a priori $PM_{10}$ simulation does not show a significant deviation from observations (Figure S2(f)), this may due to that the underestimations of $PM_{10-2.5}$ were partly compensated by the overestimated $PM_{2.5}$ concentrations over the Central region (Fig. S1(f)).

Meanwhile, we used the $PM_{2.5}/PM_{10}$ ratio to investigate the potential causes of the increases in $PM_{10}$ emission over the NW and Central regions in the revised manuscript, which is an indicator of the potential sources of particular matter. A lower $PM_{2.5}/PM_{10}$ ratio usually indicates significant contributions from natural sources such as dust (Wang et al., 2015; Fan et al., 2021). As we can see from Fig.R7, the $PM_{2.5}/PM_{10}$ ratio was stable during the P1 and P2 period, but it decreased substantially during the P3 period, from 0.81 to 0.48 over the NW region and from 0.77 to 0.53 over the Central region, which suggests larger contributions of dust emissions to the PM10 concentrations during the P3 period. Moreover, the NW and Central region are typical source areas of dust in China, therefore the increasing of $PM_{10}$ emissions over NW and Central regions may be mainly related to the enhanced dust emissions. following the suggestion of reviewer, we added more explanations to the increased $PM_{10}$ emissions over the NW and Central region in the revised manuscript (please see lines 342–348).

[Figure]

**Figure R6: Timeseries of observed and simulated PM$_{10}$ – PM$_{2.5}$ concentrations over Central region during COVID-19 pandemic.**

[Figure]

**Figure R7: Timeseries of PM$_{2.5}$/PM$_{10}$ ratio during COVID-19 pandemic over (a) NW and (b) Central region**

**Comment 5**: Line 306: east China -> southeast China

**Reply:** Done

**Comment 6**: Lines 311-312: There is a change in the values of $SO_2$ and $PM_{10}$.

**Reply:** Thanks for pointing out this mistake. The captions of Fig. 4d and Fig. 4f were wrongly labeled. We have corrected this in the revised manuscript.

**Comment 7:** Lines 313-315: The authors suggest that CO emissions decline significantly, as CO's transportation share (18%) is higher than $SO_2$(5%) and $PM_{2.5}$ (6%) (as shown in Figure 4). However, the percentage decrease in emissions is insignificant (-10.6% vs. -9.7% and -7.9%, as shown in Table 2). Furthermore, while the transportation share of $PM_{10}$ emissions is only 2%, the emission decrease is -12.1%, which is greater than that of CO. Hence, other factors beyond transportation may have influenced the reduction in anthropogenic emissions during P2. Therefore, the authors should clarify their results.

**Reply:** Thanks for raising this important issue. We agree with the review that the percentage decrease in emissions of CO, $SO_2$ and $PM_{2.5}$ is not significant compared with the differences in their transportation share. This may be on the one hand due to the uncertainty in the estimated relative contributions of different sectors to the total emissions of CO, $SO_2$ and $PM_{2.5}$, on the other hand were possibly due to the uncertainty in the emission inversions, especially considering that the decreasing trend of CO, $SO_2$ and $PM_{2.5}$ were not significant. Also, other factors beyond transportation may have influenced the reductions of anthropogenic emissions during P2 period. For example, the larger reductions of $PM_{10}$ emissions may be related in part to the reduced dust emissions due to shutting down of construction sites during the lockdown period (Li et al., 2020). Following the suggestion of reviewer, we have clarified it in the revised manuscript (please see lines 359–366).

**Comment 8:** Lines 317-318: The values given are incorrect (e.g., $SO_2$ is 77.6%, not 86%).

**Reply:** Thanks very much for your careful check. The given value is correct. It was that the caption of Fig. 4d and Fig. 4f was labeled wrongly, and we have corrected this mistake in the revised manuscript.

**Comment 9:** Lines 329-389: (section 3.3) The results presented by the authors, such as the significant contribution of meteorological fields to $PM_{2.5}$ during the pandemic and the titration effect on $O_3$, have been reported in previous studies. Hence, the authors should distinguish the difference between their results and previous studies using numerical values.

**Reply:** Thanks for this suggestion. However, it is difficult to directly compare our results with previous studies due the altered definition of meteorological contribution, different reference period that used to

quantify the meteorological contributions and different targeted region. For example, in Song et al. (2021), the reference period used to determine the meteorological contribution is the corresponding period of COVID-19 pandemic in 2019. Le et al. (2020) used the multiyear climatology as the reference period. in Wang et al. (2020) and Sulaymon et al. (2021), the MI changes of $PM_{2.5}$ concentrations were defined as the difference between the modeled concentrations in high-pollution days and those in low-pollution days under hypothetical emission reduction scenario. Zhao et al. (2020) used a similar reference period to ours to determine the MI changes but they used the outdated emission inventory.

Table R1 summarized the studies that differentiated the contributions of meteorology and emission to the $PM_{2.5}$ concentrations over Beijing and the Beijing-Tianjin-Hebei (BTH) region. Note that some studies only provided the relative changes in the modeled $PM_{2.5}$ concentrations. It shows that due to the unknown emission changes during COVID-19 pandemic, the EI changes estimated by Zhao et al. (2020) were possibly largely overestimated compared to our studies (55% versus 24.7%). Both Sulaymon et al. (2021) and Wang et al. (2020) suggested negative EI changes during COVID-19 period in Beijing. This because they presumed that the emissions were largely reduced during COVID-19 lockdown which may deviate from the real changes of emissions according to our inversion results. Meanwhile, although they used same method and reference period, their results differed largely (-2.7 versus -13.4 $\mu g/m^3$) due to the different emission reduction scenario they assumed to represent the emissions during COVID-19 pandemic. Le et al. (2020) only considered the emission reductions of $NO_x$ in their sensitivity simulations without considerations of other species, therefore their calculated EI changes may be underestimated compared to our results (almost 0% versus 24.7%). However, the calculated MI changes were consistent between our study and Le et al. (2020). In terms of $O_3$, the calculated EI changes by our study were also higher than that calculated by Zhao et al. (2020) in Beijing (85.7% versus 70%). These results suggested that the EI and MI changes calculated by our study could be more reasonable, as the emissions of different species were well constrained which could better represent the temporal variation and spatial heterogeneity of emission changes during COVID-19. Following the suggestions of reviewer, we have added the comparison of our results with previous studies in the revised manuscript (please see lines 442–464)

**Table R1. calculated MI and EI changes in PM$_{2.5}$ concentrations during COVID-19 pandemic by previous studies**

| | MI changes | EI changes | Region | Reference period | Method | Reference |
|---|---|---|---|---|---|---|
| 1 | 26.79 µg/m$^3$ | -21.84 µg/m$^3$ | Beijing | January 23-March 10, 2019 versus January 23-March 10, 2020 | observation-based wind-decomposition method | Song et al. (2021) |
| 2 | Around 20 µg/m$^3$ | -2.7 µg/m$^3$ | Beijing | January 01 to February 29, 2020 | CTM with hypothetical emission reduction scenario | Sulaymon et al. (2021) |
| 3 | Around 45 µg/m$^3$ | -13.4 µg/m$^3$ | Beijing | January 01 to February 29, 2020 | CTM with hypothetical emission reduction scenario | Wang et al. (2020) |
| 4 | 31.3% | Around 0% | Beijing-Tianjin-Hebei | January 01 to February 13, 2020 | CTM sensitivity simulations using different emission rates and multiyear climatology | Le et al. (2020) |
| 5 | Around 5% | Around 55% | Beijing | January 16-22, 2020 versus January 26 to February 1, 2020 | CTM with fixed emission inventory for 2017 | Zhao et al. (2020) |
| 6 | 17.5 µg/m$^3$ (34.0%) | 12.7 µg/m$^3$ (24.7%) | Beijing | January 1-20, 2020 versus January 21 to February 9, 2020 | CTM with inversion emission inventory | This study |

**Comment 9:** Lines 340-341: The relative overestimation of ozone is not clear. Please provide a specific value. Also, Figure 7 is not related to this.

**Reply:** Thanks for this comment. The simulated increases in O$_3$ concentrations from pre-lockdown to lockdown period were 30.9 µg/m$^3$ over the NCP region, which is slightly higher than the observed increases in O$_3$ concentrations (28.3 µg/m$^3$). Following the suggestions of reviewer, we have clarified in the revised manuscript (please see lines 389–390), and we feel sorry for the wrong quotation of "Fig. 7" there which has been corrected in the revised manuscript (please see lines 390).

**Comment 10:** Lines 359-361: The author's assertion that the rise in PM$_{2.5}$ levels in the Beijing region is mainly due to fireworks during the Spring Festival is not supported by sufficient evidence, as there is no evidence that the increase in fireworks emissions is unique to Beijing.

**Reply:** Thanks for this important comment. Following the suggestions of reviewer, we have added more explanation of the possible causes of the PM$_{2.5}$ emission increases in the Beijing during the lockdown period through literature review and analysis of the PM$_{2.5}$ compositions. Zuo et al. (2022) analyzed the

variations in the PM$_{2.5}$ sources based on the measurement of stable Cu and Si isotopic signature and metal concentrations of PM$_{2.5}$ in Beijing, which indicated that the primary PM$_{2.5}$ emissions did not decrease in Beijing during COVID-19 lockdown, and that the PM-associated industrial emissions may increase in Beijing and its upwind region during the lockdown period. Meanwhile, substantial high levels of potassium (K) and barium (Mg) were observed over Beijing during Spring Festival as seen from Fig. R8, which is an important fingerprint of the firework emissions. This suggest that the emissions from fireworks during Spring Festival were also a potential contributor to the increased of PM$_{2.5}$ emissions in Beijing, which is consistent with the measurement by Ma et al. (2022) and Dai et al. (2020). Therefore, the increased PM$_{2.5}$ emissions during lockdown period in Beijing may be attributed to the increased industrial PM$_{2.5}$ emissions and the firework emissions, which compensated the emission reductions from the traffic emissions. Following the suggestions of reviewer, we have clarified this in the revised manuscript (please see lines 416–418)

[Figure]

**Figure R8: Timeseries of averaged concentrations of potassium and magnesium ion during COVID-19 pandemic over the Beijing.**

**Comment 11:** Line 441: Some subscripts are misspelled.

**Reply:** We have corrected it in the revised manuscript.

**Comment 12:** Lines 460-461: Is the unit for ozone also μg m$^{-3}$?

**Reply:** Yes, the unit for ozone is also μg/m$^3$.

**Comment 13:** Line S36: Figure S4 -> Figure S7

**Reply:** We have corrected it in the revised manuscript.

**References**

Dai, Q., Liu, B., Bi, X., Wu, J., Liang, D., Zhang, Y., Feng, Y., and Hopke, P. K.: Dispersion Normalized PMF Provides Insights into the Significant Changes in Source Contributions to PM2.5 after the COVID-19 Outbreak, Environ. Sci. Technol., 54, 9917-9927, https://doi.org/10.1021/acs.est.0c02776, 2020.

Fan, H., Zhao, C., Yang, Y., and Yang, X.: Spatio-Temporal Variations of the PM2.5/PM10 Ratios and Its Application to Air Pollution Type Classification in China, Front. Environ. Sci., 9, https://doi.org/10.3389/fenvs.2021.692440, 2021.

Le, T. H., Wang, Y., Liu, L., Yang, J. N., Yung, Y. L., Li, G. H., and Seinfeld, J. H.: Unexpected air pollution with marked emission reductions during the COVID-19 outbreak in China, Science, 369, 702-+, https://doi.org/10.1126/science.abb7431, 2020.

Li, L., Li, Q., Huang, L., Wang, Q., Zhu, A., Xu, J., Liu, Z., Li, H., Shi, L., Li, R., Azari, M., Wang, Y., Zhang, X., Liu, Z., Zhu, Y., Zhang, K., Xue, S., Ooi, M. C. G., Zhang, D., and Chan, A.: Air quality changes during the COVID-19 lockdown over the Yangtze River Delta Region: An insight into the impact of human activity pattern changes on air pollution variation, Sci. Total Environ., 732, 139282, https://doi.org/10.1016/j.scitotenv.2020.139282, 2020.

Ma, T., Duan, F. K., Ma, Y. L., Zhang, Q. Q., Xu, Y. Z., Li, W. G., Zhu, L. D., and He, K. B.: Unbalanced emission reductions and adverse meteorological conditions facilitate the formation of secondary pollutants during the COVID-19 lockdown in Beijing, Sci. Total Environ., 838, 8, https://doi.org/10.1016/j.scitotenv.2022.155970, 2022.

Song, Y. S., Lin, C. Q., Li, Y., Lau, A. K. H., Fung, J. C. H., Lu, X. C., Guo, C., Ma, J., and Lao, X. Q.: An improved decomposition method to differentiate meteorological and anthropogenic effects on air pollution: A national study in China during the COVID-19 lockdown period, Atmos. Environ., 250, 9, https://doi.org/10.1016/j.atmosenv.2021.118270, 2021.

Sulaymon, I. D., Zhang, Y., Hopke, P. K., Hu, J., Zhang, Y., Li, L., Mei, X., Gong, K., Shi, Z., Zhao, B., and Zhao, F.: Persistent high PM2.5 pollution driven by unfavorable meteorological conditions during the COVID-19 lockdown period in the Beijing-Tianjin-Hebei region, China, Environ. Res., 198, 111186, https://doi.org/10.1016/j.envres.2021.111186, 2021.

Wang, P., Chen, K., Zhu, S., Wang, P., and Zhang, H.: Severe air pollution events not avoided by reduced anthropogenic activities during COVID-19 outbreak, Resources, Conservation and Recycling, 158, 104814, https://doi.org/10.1016/j.resconrec.2020.104814, 2020.

Wang, Y. Q., Zhang, X. Y., Sun, J. Y., Zhang, X. C., Che, H. Z., and Li, Y.: Spatial and temporal variations of the concentrations of PM$_{10}$, PM$_{2.5}$ and PM$_1$ in China, Atmos. Chem. Phys., 15, 13585-13598, https://doi.org/10.5194/acp-15-13585-2015, 2015.

Zhao, Y. B., Zhang, K., Xu, X. T., Shen, H. Z., Zhu, X., Zhang, Y. X., Hu, Y. T., and Shen, G. F.: Substantial Changes in Nitrogen Dioxide and Ozone after Excluding Meteorological Impacts during the COVID-19 Outbreak in Mainland China, Environ. Sci. Technol. Lett., 7, 402-408, https://doi.org/10.1021/acs.estlett.0c00304, 2020.

Zuo, P. J., Zong, Z., Zheng, B., Bi, J. Z., Zhang, Q. H., Li, W., Zhang, J. W., Yang, X. Z., Chen, Z. G., Yang, H., Lu, D. W., Zhang, Q. H., Liu, Q., and Jiang, G. B.: New Insights into Unexpected Severe PM2.5 Pollution during the SARS and COVID-19 Pandemic Periods in Beijing, Environ. Sci. Technol., 56, 155-164, https://doi.org/10.1021/acs.est.1c05383, 2022.